# Impact of Diverse Dietary Patterns on Cognitive Health: Cumulative Evidence from Prospective Cohort Studies

**DOI:** 10.3390/nu17213469

**Published:** 2025-11-03

**Authors:** Youngyo Kim, Minkyung Je, Kyeonghoon Kang, Yoona Kim

**Affiliations:** 1Department of Food and Nutrition, Research Institute of Molecular Alchemy, Gyeongsang National University, Jinju 52828, Republic of Korea; youngyokim@gnu.ac.kr; 2Department of Food and Nutrition, Gyeongsang National University, Jinju 52828, Republic of Korea

**Keywords:** prospective studies, Mediterranean-dietary approaches to stop hypertension intervention for neurodegenerative delay diet, healthy plant-based dietary index, cognitive outcomes

## Abstract

Background/Objectives: The aging population is associated with an increased incidence of dementia, which deteriorates the quality of life of adults, leading to an elevated socioeconomic burden. This review aimed to extensively examine which dietary patterns favorably influence cognitive outcomes based on prospective cohort studies of adults. Methods: A literature search was performed in the PubMed^®^/MEDLINE^®^ database up to 30 October 2024. Results: One hundred and eighteen publications were included. In a comparison of high and low categories, the Mediterranean-Dietary Approaches to Stop Hypertension Intervention for Neurodegenerative Delay (MIND) diet increased cognitive function and reduced the risk of cognitive impairment. The Mediterranean (MED) diet improved cognitive function but showed no beneficial effects on cognitive impairment or dementia. The Dietary Approaches to Stop Hypertension (DASH) diet lowered the risk of cognitive impairment but showed no alteration in either cognitive function or dementia. The Healthy Eating Index (HEI) did not alter cognitive function, cognitive impairment, or dementia. The healthy plant-based dietary index (hPDI) decreased the risk of cognitive impairment and dementia, whereas the unhealthy plant-based dietary index (uPDI) elevated the risk of cognitive impairment. The Western dietary pattern (WDP) decreased cognitive function. Conclusions: The MIND diet and hPDI appeared to be effective, while the WDP appeared to be detrimental for cognitive health. Further prospective cohort studies of healthy and unhealthy dietary patterns are required to confirm the association between diverse diets and cognitive health.

## 1. Introduction

As a result of advances in medical technology and improvements in nutrition and hygiene, life expectancy is increasing worldwide, leading to an aging population. Aging changes various body functions, causing a decline in cognitive function due to changes in the brain [1]. Dementia is a decline in cognitive function that causes impairments that interfere with independent living in daily life [2]. Globally, the number of people with dementia is expected to increase rapidly from 57.4 million in 2019 to 152.8 million in 2050 [3]. Dementia not only reduces the quality of life of individuals but also increases healthcare costs for families, communities, and governments, creating a global burden. Risk factors for dementia include various factors such as education level, disease, air pollution, and lifestyle, and controlling these risk factors can be a way to prevent dementia [4]. Risk factors for the disease include non-modifiable factors such as age and modifiable factors such as lifestyle habits. Social factors such as low education level or social isolation may increase the risk of dementia [5,6]. Obesity, diabetes, high blood pressure, high low-density lipoprotein (LDL) cholesterol, and depression have also been shown to raise the risk of dementia [4]. Neglecting vision and hearing loss, brain damage, and air pollution have also been shown to increase the risk of dementia [4,7,8,9]. In addition, it is known that unhealthy lifestyle habits such as smoking, excessive alcohol consumption, and lack of physical activity can have a negative impact on cognitive function [4]. Diet, as one of the lifestyle habits, is classified as a modifiable risk factor for dementia [10].

Many studies have been conducted on the association between food or nutrient intake and decline in cognitive function. However, recently, there has been an increase in research on the association between dietary patterns and cognitive function [11]. There are several advantages to considering dietary patterns rather than single foods or nutrients as exposure factors in studying the relationship between dietary intake and disease. First, dietary patterns can help us identify synergistic interactions across food combinations [12]. Second, dietary patterns better reflect an individual’s daily eating habits than individual foods or nutrients. Third, because people typically eat various foods in different combinations, dietary patterns are more useful in providing dietary guidelines than individual foods or nutrients [13]. Therefore, examining the association between dietary patterns and cognitive decline may help develop dementia prevention strategies. Although there have been attempts to integrate the results of studies on the association between various dietary patterns and cognitive decline and dementia, in the past 2–3 years, a significant number of additional studies have been reported on dietary patterns and the risk of cognitive decline. Therefore, we conducted a comprehensive review of prospective cohort studies to determine the association between various dietary patterns and cognitive decline and dementia and to obtain updated, integrated results. This review aimed to understand the differences in cognitive function and dementia incidence across different dietary patterns and to explore dietary patterns that might help prevent cognitive decline.

## 2. Materials and Methods

### 2.1. Information Sources and Search Strategy

A literature search was carried out for manuscripts published up to 30 October 2024 in the PubMed^®^/MEDLINE^®^ database (https://pubmed.ncbi.nlm.nih.gov/pubmed/ (accessed on 30 October 2024)).

The search strategy for this review is presented in Table 1.

The search strategy consisted of prospective studies combined with dietary patterns, including the Mediterranean-Dietary Approaches to Stop Hypertension Intervention for Neurodegenerative Delay (MIND) diet, the Mediterranean (MED) diet, the Dietary Approaches to Stop Hypertension (DASH), the Healthy Eating Index (HEI), the Alternative Healthy Eating Index (AHEI), plant-based diet, the ketogenic diet, healthy or unhealthy dietary patterns, and other relevant terms. The search for publications was restricted to full texts available in the English language. Additionally, the reference lists of the included studies were manually reviewed to ensure that all relevant publications were included.

### 2.2. Study Selection and Eligibility Criteria

The inclusion and exclusion criteria for study selection in this review are presented in Table 2.

Study selection and exclusion were performed using EndNote software (version X9, Clarivate Analytics, Philadelphia, PA, USA). The duplicated publications were removed using the software tool.

First, three researchers (Y.K., M.J., and K.K.) independently reviewed the titles and abstracts of all obtained literature. Then, a review of the full texts obtained was performed for eligibility according to the following criteria: (1) Publications with a human observational prospective study. Publications with other study types, such as interventions, randomized controlled clinical trials (RCTs), reviews, meta-analyses, editorials, commentaries, letters, or cross-sectional studies, were excluded. (2) Publications assessing the effects of different dietary patterns on cognitive outcomes (cognitive function, cognitive impairment, and dementia). Publications with unrelated exposures (e.g., food consumption, medicine) or outcomes were excluded. (3) Publications with study subjects included without selection. Some study subjects with underlying medical conditions were excluded.

Figure 1 shows a flow diagram of the literature search for this review.

### 2.3. Data Extraction

We extracted data from publications that met the eligibility criteria. The following variables were used: dietary pattern, first author, year of publication, country, study name, adherence to dietary pattern, sample size, percentage of female subjects, age (range or mean age), follow-up period, and outcomes.

### 2.4. Quality Assessment

Three independent researchers (Y.K., M.J., and K.K.) evaluated the quality of the prospective studies included in this review. The tools utilized for quality assessment were derived from the Newcastle–Ottawa Scale [14]. In this review, the scale consisted of three domains: selection of the study population (0 to 4 points), comparability for controlling confounders (0 to 2 points), and outcome ascertainment (0 to 3 points). The maximum score for quality assessment was 9 points, which was categorized as low (0 to 3 points), moderate (4 to 6 points), or high quality (7 to 9 points), respectively. Disagreements were resolved through discussion of the original papers to ensure accurate quality assessment.

## 3. Results

### 3.1. Study Selection

This review identified a total of 897 eligible publications, which included 893 from a database search and 4 from a manual search. After screening the titles and abstracts, 682 publications were initially excluded due to their publication type, with 215 publications remaining. Of these, 97 publications were excluded during the full-text review: 48 publications were excluded due to their unrelated exposures, and 32 publications due to unrelated outcomes. Furthermore, 7 publications including non-adult populations and 10 publications with participants having underlying medical conditions were excluded. Finally, 118 publications were included in this review (Figure 1).

### 3.2. Overview of Study Characteristics

The characteristics of the 118 publications [15,16,17,18,19,20,21,22,23,24,25,26,27,28,29,30,31,32,33,34,35,36,37,38,39,40,41,42,43,44,45,46,47,48,49,50,51,52,53,54,55,56,57,58,59,60,61,62,63,64,65,66,67,68,69,70,71,72,73,74,75,76,77,78,79,80,81,82,83,84,85,86,87,88,89,90,91,92,93,94,95,96,97,98,99,100,101,102,103,104,105,106,107,108,109,110,111,112,113,114,115,116,117,118,119,120,121,122,123,124,125,126,127,128,129,130,131,132] are presented in Table 3, Table 4, Table 5, Table 6, Table 7, Table 8, Table 9 and Table 10. In these 118 publications [15,16,17,18,19,20,21,22,23,24,25,26,27,28,29,30,31,32,33,34,35,36,37,38,39,40,41,42,43,44,45,46,47,48,49,50,51,52,53,54,55,56,57,58,59,60,61,62,63,64,65,66,67,68,69,70,71,72,73,74,75,76,77,78,79,80,81,82,83,84,85,86,87,88,89,90,91,92,93,94,95,96,97,98,99,100,101,102,103,104,105,106,107,108,109,110,111,112,113,114,115,116,117,118,119,120,121,122,123,124,125,126,127,128,129,130,131,132], numerous dietary patterns were investigated to measure their influence on cognition: MIND (*n* = 28 publications [15,16,17,18,19,20,21,22,23,24,25,26,27,28,29,30,31,32,33,34,35,36,37,38,39,40,41,42]), MED (*n* = 55 publications [19,20,25,26,29,36,37,38,41,43,44,45,46,47,48,49,50,51,52,53,54,55,56,57,58,59,60,61,62,63,64,65,66,67,68,69,70,71,72,73,74,75,76,77,78,79,80,81,82,83,84,85,86,87,88]), DASH (*n* = 17 publications [19,20,36,37,41,49,50,60,63,64,68,69,76,80,89,90,91]), HEI/AHEI (*n* = 11 publications [30,37,50,60,63,64,67,69,85,92,93]), plant-based dietary index (PDI)/healthy plant-based dietary index (hPDI)/unhealthy plant-based dietary index (uPDI) (*n* = 8 publications [26,64,94,95,96,97,98,99]), healthy dietary patterns (*n* = 33 publications [16,29,37,41,52,57,70,72,82,92,100,101,102,103,104,105,106,107,108,109,110,111,112,113,114,115,116,117,118,119,120,121,122]), Western dietary pattern (WDP) (*n* = 12 publications [51,72,92,100,101,102,103,104,105,123,124,125]), and other dietary patterns (*n* = 13 publications [55,82,104,109,114,115,126,127,128,129,130,131,132]).

Thirty-nine publications [16,19,20,25,26,29,30,36,37,38,41,49,50,51,52,55,57,60,63,64,67,68,69,70,72,76,80,82,85,92,100,101,102,103,104,105,109,114,115] conducted an investigation into the effects of two or more dietary patterns on cognition. In our review, the tables were classified according to the prevailing dietary patterns in order to ascertain the effect of each dietary pattern on cognitive function in adults.

Of the 118 publications included in this review, 44 publications investigated cohorts conducted in North America, while the other publications investigated cohorts from Europe (*n* = 43), Asia (*n* = 23), and Australia (*n* = 8). In addition, some cohorts were used multiple times, as follows: United Kingdom (UK) Biobank study (*n* = 6) [26,30,44,96,127,128], Chicago Health and Aging Project (CHAP) (*n* = 5) [18,31,56,85,97], China Health and Nutrition Survey (CHNS) (*n* = 5) [28,75,118,119,129], Rush Memory and Aging Project (MAP) (*n* = 5) [15,31,34,42,76], Nurses’ Health Study (NHS) (*n* = 5) [40,50,77,91,123], Chinese Longitudinal Healthy Longevity Surveys (CLHLS) (*n* = 4) [24,98,99,126], Quebec Longitudinal Study on Nutrition and Successful Aging (NuAge) study (*n* = 4) [93,102,105,124], Health and Retirement Study (HRS) (*n* = 3) [19,20,27], Personality and Total Health Through Life Cohort (PATH) study (*n* = 3) [16,38,84], REasons for Geographic and Racial Differences in Stroke (REGARDS) (*n* = 3) [17,20,79], Seguimiento Universidad de Navarra (SUN) cohort study (*n* = 3) [37,51,71], Swedish National study on Aging and Care in Kungsholmen (SNAC-K) (*n* = 3) [41,103,106], Whitehall II study (WII) (*n* = 3) [27,92,132], Hellenic Epidemiological Longitudinal Investigation of Aging and Diet (HELIAD) study (*n* = 3) [45,47,58], Framingham Heart Study (FHS) Offspring cohort (*n* = 2) [21,27], Rotterdam study (*n* = 2) [29,94], Women’s Health Initiative Memory Study (WHIMS) (*n* = 2) [31,69], Three-City (3C) Bordeaux study (*n* = 2) [32,86], Boston Puerto Rican Health Study (BPRHS) (*n* = 2) [33,63], Sydney Memory and Ageing Study (MAS) (*n* = 2) [49,110], Atherosclerosis Risk in Communities (ARIC) study (*n* = 2) [60,125], Singapore Chinese Health Study (SCHS) (*n* = 2) [64,90], European Prospective Investigation into Cancer and Nutrition (EPIC)-Greece study (*n* = 2) [73,88], and Ohsaki cohort (*n* = 2) [109,113].

In this review, the quality assessment results of 192 studies (118 publications) are presented in Table 3, Table 4, Table 5, Table 6, Table 7, Table 8, Table 9 and Table 10. The overall quality assessment results showed high quality, with an average score of 7.4 (high quality). Of these, 153 studies (92 publications) were rated as high quality and 39 studies (26 publications) were rated as moderate quality.

### 3.3. Dietary Patterns

#### 3.3.1. Mediterranean-Dietary Approaches to Stop Hypertension Intervention for Neurodegenerative Delay (MIND) Diet

Table 3 summarizes the associations between the MIND diet and cognitive outcomes in prospective studies. A total of 36 prospective studies (28 publications [15,16,17,18,19,20,21,22,23,24,25,26,27,28,29,30,31,32,33,34,35,36,37,38,39,40,41,42]) investigated the effect of the MIND diet on cognition (Table 3).

The studies in the 28 publications included in this review were conducted in the United States of America (USA) (*n* = 13), Europe (*n* = 10), Australia (*n* = 2), and Asia (*n* = 3). The cohort studies included in this review are as follows: Religious Orders Study (ROS) [15], MAP [15,31,34,42], PATH study [16,38], REGARDS [17,20], CHAP [18,31], HRS [19,27], FHS Offspring cohort [21,27], Long-Term Research Grant Scheme—Towards Useful Aging (LRGS-TUA) and Fundamental Research Grant Scheme (FRGS) [22], VitaminD3–Omega3–Home Exercise–Healthy Ageing and Longevity Trial (DO-HEALTH) clinical trial [23], CLHLS [24], United Kingdom Adult Twin Registry (TwinsUK) [25], UK Biobank study [26,30], WII [27], CHNS [28], Rotterdam study [29], WHIMS [31], 3C Bordeaux study [32], BPRHS [33], FHS [35], PREvención con DIeta MEDiterránea (PREDIMED)-Plus trial [36], SUN cohort study [37], NutriNet-Santé study [39], NHS [40], and SNAC-K [41]. The sample size of the 36 studies was 366,762 subjects (range: 220 to 120,661), with the mean age ranging from 51.9 ± 12.5 to 82.5 ± 6.0 years. The follow-up duration ranged from 2 to 20 years.

In 36 studies (28 publications [15,16,17,18,19,20,21,22,23,24,25,26,27,28,29,30,31,32,33,34,35,36,37,38,39,40,41,42]), the quality assessment results showed a mean quality assessment score of 7.3 (high quality), with 29 studies (21 publications [15,16,17,18,20,24,26,27,28,29,30,31,32,33,35,36,37,38,39,40,41]) scoring high on quality and 7 studies (7 publications [19,21,22,23,25,34,42]) scoring moderate.

The MIND diet significantly increased cognitive function in 9 (9 publications [18,25,28,30,31,33,34,41,42]) of 17 studies (15 publications [18,19,25,28,30,31,33,34,35,36,37,39,40,41,42]). In addition, significant decreases in the risk of cognitive impairment were found with higher adherence to the MIND diet in 7 (7 publications [15,16,17,20,22,24,38]) of 10 studies (8 publications [15,16,17,20,22,23,24,38]). However, 8 (5 publications [15,21,29,31,32]) of 16 studies (8 publications [15,21,26,27,29,30,31,32]) reported that the MIND diet effectively reduced the risk of dementia, while the other 8 studies (5 publications [26,27,29,30,31]) found no significant effect.

Li et al. (2024) [15] investigated the association between the MIND diet and the risk of dementia, linking the brain transcriptomic profile in the ROS and MAP. Fifty genes were associated with the MIND diet score. In all subjects with ribonucleic acid sequencing (RNA-Seq) data (*n* = 1204) and an independent set of subjects with RNA-Seq data (*n* = 722), a significant association between the MIND diet score and reduced risk of dementia was observed, while no significant association between the MIND diet score and reduced risk of mild cognitive impairment (MCI) was found. In subjects with dietary and RNA-Seq data (*n* = 444), the MIND diet score was associated with reduced risk of MCI, as well as reduced risk of dementia [15].

Thomas et al. (2024) [21] observed that the highest MIND score was associated with a lower risk of dementia over 14 years of follow-up in the FHS Offspring Cohort study of 1644 individuals aged over 60 years.

Chen et al. (2023) [27] examined the association between the MIND diet and incidence of dementia in 3 prospective studies: the WII study, the HRS, and FHS Offspring cohort. Among 775 participants who developed incident all-cause dementia (220 of 105,949 person-years in the WII study, 338 of 28,934 person-years in the HRS, and 217 of 31,633 person-years in the FHS Offspring cohort), none of the prospective studies showed a significant association between the MIND diet and the risk of dementia. When the 3 prospective studies were pooled, the hazard ratios (HRs) were 0.81 [95% confidence interval (CI) = 0.67, 0.98] for the highest vs. lowest tertiles of the MIND score and 0.83 (95% CI = 0.72, 0.95; *p* for trend = 0.01) for every 3-point increment in the multivariable-adjusted model [27].

de Crom et al. (2022) [29] calculated the MIND diet score from validated food frequency questionnaires from baseline I (1989–1993) and baseline II (2009–2013) of the population-based Rotterdam Study. They found no association between the MIND diet score and the risk of dementia during the mean follow-up period of 15.6 years from baseline I, while the greater MIND diet score was associated with a lower risk of dementia during the mean follow-up period of 5.9 years from baseline II [29].

#### 3.3.2. Mediterranean (MED) Diet

Table 4 summarizes the associations between the MED diet and cognitive outcomes in prospective studies. The effects of the MED diet on cognition were observed in 56 prospective studies (55 publications [19,20,25,26,29,36,37,38,41,43,44,45,46,47,48,49,50,51,52,53,54,55,56,57,58,59,60,61,62,63,64,65,66,67,68,69,70,71,72,73,74,75,76,77,78,79,80,81,82,83,84,85,86,87,88]) (Table 4).

The studies in the 55 publications included in this review were conducted in Europe (*n* = 27), the USA (*n* = 21), Australia (*n* = 4), and Asia (*n* = 3). The cohort studies included in this review are as follows: HRS [19], REGARDS [20,79], TwinsUK [25], UK Biobank study [26,44], Rotterdam study [29], HELIAD study [45,47,58], European Prevention of Alzheimer’s Dementia Longitudinal Cohort Study (EPAD LCS) [46], Hispanic Community Health Study/Study of Latinos (HCHS/SOL) study of Latinos—Investigation of Neurocognitive Aging (SOL–INCA) [48], MAS [49], NHS [50,77], SUN cohort study [37,51,71], Malmö Diet and Cancer study (MDCS) [52], Maine–Syracuse Longitudinal Study (MSLS) [53], Monzino 80-plus study [54], Lothian Birth Cohort 1936 study [55], CHAP [56,85], Doetinchem Cohort Study [57], EPIC-Spain Dementia Cohort study [59], ARIC study [60], Age-Related Eye Disease Study (AREDS) and AREDS2 [61], PATH study [38,84], EPIC-Norfolk Study [62], BPRHS [63], SCHS [64], SNAC-K [41], Invecchiare in Chianti, aging in the Chianti area (InCHIANTI) study [65], Health Professionals’ Follow-up Study (HPFS) [66], Rancho Bernardo Study (RBS) of Healthy Aging study [67], Swedish Infrastructure for Medical Population-based Life-course Environmental Research, previously the Swedish Mammography Cohort and the Cohort of Swedish Men (SIMPLER) study [68], WHIMS [69], Uppsala longitudinal study [70], Australian Imaging, Biomarkers and Lifestyle study of ageing (AIBL) study [72], EPIC-Greece study [73,88], Health, Aging, and Body Composition (Health ABC) study [74], CHNS [75], MAP [76], Women’s Health Study [78], Cache County Memory Study (CCMS) [80], Supplementation with Vitamins and Mineral Antioxidants (SU.VI.MAX) study [81], Prospective Investigation of the Vasculature in Uppsala Seniors (PIVUS) [82], Women’s Antioxidant Cardiovascular Study (WACS) [83], 3C study [86], Washington Heights–Inwood Columbia Aging Project (WHICAP) study [87], and two studies not available (NA) [36,43]. The sample size of the 57 studies was 530,570 subjects (range: 194 to 114,684) aged 30 to 92 years, with follow-up periods ranging from 2 to 20 years.

In 56 studies (55 publications [19,20,25,26,29,36,37,38,41,43,44,45,46,47,48,49,50,51,52,53,54,55,56,57,58,59,60,61,62,63,64,65,66,67,68,69,70,71,72,73,74,75,76,77,78,79,80,81,82,83,84,85,86,87,88]), the quality assessment results showed a mean quality assessment score of 7.6 (high quality), with 48 studies (47 publications [20,26,29,36,37,38,41,43,44,45,46,47,48,49,50,51,52,53,54,55,56,57,58,59,60,61,62,63,64,65,66,67,68,69,70,73,74,75,77,79,80,81,82,85,86,87,88]) scoring high on quality and 8 studies (8 publications [19,25,71,72,76,78,83,84]) scoring moderate.

In total, 28 (28 publications [19,25,36,41,43,45,46,47,50,55,56,57,62,63,65,66,67,71,72,73,74,75,76,77,80,81,85,86]) of 38 studies (38 publications [19,25,36,37,41,43,45,46,47,48,49,50,51,53,55,56,57,61,62,63,65,66,67,71,72,73,74,75,76,77,78,80,81,82,83,85,86,88]) found that the MED diet significantly increased cognitive function.

Yuan et al. (2022) [50] investigated the effects of the MED diet adherence on cognitive function for a long-term follow-up of 31 years in the NHS of 49,493 middle-aged women. They found that long-term adherence to the MED diet was associated with improved cognitive function in women [50].

However, no significant difference in the risk of cognitive impairment in 6 (6 publications [20,38,69,70,84,87]) of 9 studies (9 publications [20,38,61,64,69,70,79,84,87]) and of dementia in 10 (10 publications [26,29,52,54,60,68,69,70,84,86]) of 16 studies (14 publications [26,29,44,47,52,54,58,59,60,68,69,70,84,86]) were found.

Two research teams investigated the association between MED diet adherence and the risk of dementia using the UK Biobank prospective cohort study [26,44]. Zhang et al. (2023) [26] found no significant association between the MED diet score (MDS) and the risk of dementia during the mean follow-up period of 9.4 years when analyzing a large sample of 114,684 subjects aged over 50 years from the UK Biobank prospective cohort study. On the other hand, Shannon et al. (2023) [44] showed a significant association between a higher MED diet intake and lower risk of dementia during the mean follow-up period of 9.1 years when analyzing 60,298 subjects aged over 50 years from the UK Biobank prospective cohort study.

#### 3.3.3. Dietary Approaches to Stop Hypertension (DASH) Diet

Table 5 summarizes the associations between the DASH diet and cognitive outcomes in prospective studies. The effect of the DASH diet on cognition was observed in 17 studies (17 publications [19,20,36,37,41,49,50,60,63,64,68,69,76,80,89,90,91]) (Table 5).

The cohort studies included in this review are as follows: HRS [19], REGARDS [20], MAS [49], NHS [50,91], Multi-Ethnic Study of Atherosclerosis (MESA) cohort study [89], SCHS [64,90], SUN cohort study [37], ARIC study [60], BPRHS [63], SNAC-K [41], SIMPLER study [68], WHIMS [69], MAP [76], CCMS [80], and NA [36]. The sample size of 17 studies was 182,475 subjects (range: 557 to 49,493), with the mean age ranging from 48 ± 7 to 81.5 ± 7.1 years. The follow-up duration ranged from 2 to 27 years.

In 17 studies (17 publications [19,20,36,37,41,49,50,60,63,64,68,69,76,80,89,90,91]), the quality assessment results showed a mean quality assessment score of 7.2 (high quality), with 13 studies (13 publications [20,36,37,41,49,50,60,63,64,68,69,80,90]) scoring high on quality and 4 studies (4 publications [19,76,89,91]) scoring moderate.

No significant association between the DASH diet and cognitive function in 5 (5 publications [36,37,41,49,89]) of 11 studies (11 publications [19,36,37,41,49,50,63,76,80,89,91]) or between the DASH diet and dementia in 3 studies (3 publications [60,68,69]) was found. The three (3 publications [20,64,90]) of four cohorts (4 publications [20,64,69,90]) found that the DASH diet significantly decreased the risk of cognitive impairment in adult subjects.

Tong et al. (2021) [90] showed that a higher DASH diet score was significantly associated with a lower risk of cognitive impairment in a dose-dependent manner during the mean follow-up period of 3 years from the cohort data of SCHS. Wengreen et al. (2013) [80] found a significant association between higher DASH diet scores and improved cognitive function during the mean follow-up period of 11 years in the CCMS of 716 subjects. Hu et al. (2020) [60] found no association between the DASH score and the risk of dementia in 13,630 adults from the ARIC Study.

#### 3.3.4. Healthy Eating Index (HEI)

Table 6 summarizes the associations between the HEI and cognitive outcomes in prospective studies. A total of 11 prospective studies (11 publications [30,37,50,60,63,64,67,69,85,92,93]) investigated the association between the HEI and cognition (Table 6).

The cohort studies included in this review are as follows: UK Biobank study [30], NHS [50], SUN cohort study [37], ARIC study [60], BPRHS [63], SCHS [64], WII [92], RBS of Healthy Aging study [67], WHIMS [69], NuAge study [93], and CHAP [85]. The sample size of the 36 studies was 223,522 subjects (range: 557 to 120,661), with the mean age ranging from 48 ± 7 to 75.4 ± 6.2 years. The follow-up duration ranged from 2 to 31 years.

In 11 studies (11 publications [30,37,50,60,63,64,67,69,85,92,93]), the quality assessment results showed a mean quality assessment score of 7.4 (high quality), with 10 studies (10 publications [30,37,50,60,63,64,67,69,85,92]) scoring high on quality and 1 study (1 publication [93]) scoring moderate.

No significant association was found between the HEI or AHEI and cognitive function in 4 (4 publications [67,85,92,93]) of 9 studies (8 publications [30,37,50,63,67,85,92,93]), cognitive impairment in 1 [69] of 2 studies [64,69], and dementia in 4 (4 publications [30,60,69,92]) of 5 studies (4 publications [30,60,69,92]).

Cornelis et al. (2022) [30] showed a significant association between the higher AHEI-2010 scores and improved cognitive function but no significant association between the higher AHEI-2010 scores and the risk of dementia during the mean follow-up period of 10.5 years when analyzing 120,661 subjects aged over 50 years from the UK Biobank prospective cohort study.

Mattei et al. (2019) [63] showed that a higher HEI-2005 score was significantly associated with improved memory function and word recognition in individuals without type 2 diabetes during the mean follow-up period of 2 years from the longitudinal BPRHS. Hu et al. (2020) [60] found no association between the AHEI-2010 score and the risk of dementia in 13,630 adults from the ARIC Study.

#### 3.3.5. Plant-Based Dietary Pattern

Table 7 summarizes the associations between plant-based patterns and cognitive outcomes in prospective studies. The effects of the plant-based pattern diet on cognition were observed in 8 prospective studies (8 publications [26,64,94,95,96,97,98,99]). The adherence of plant-based patterns was categorized as PDI, hPDI, and uPDI. The effect of each pattern varies with adherence; thus, we observed the results of the 3 plant-based patterns (Table 7).

Of 8 studies (8 publications [26,64,94,95,96,97,98,99]), the hPDI was investigated in 8 studies, and the PDI and uPDI were investigated in 5 studies. The cohort studies included in this review are as follows: UK Biobank study [26,96], Rotterdam study [94], the B-vitamins for the Prevention of Osteoporotic Fractures (B-proof) [95], CHAP [97], CLHLS [98,99], and SCHS [64]. The sample size of the 8 studies was 336,286 subjects (range: 314 to 180,532), with the mean age ranging from 53.5 ± 6.2 to 80 ± 9.83 years. The follow-up duration ranged from 2 to 19.7 years.

In the 8 studies (8 publications [26,64,94,95,96,97,98,99]), the quality assessment results showed a mean quality assessment score of 7.9 (high quality), with 7 studies (7 publications [26,64,94,96,97,98,99]) scoring high on quality and 1 study (1 publication [95]) scoring moderate.

Adherence to the PDI [97], 1 [95] of the 2 hPDIs [95,97], and both of the 2 uPDIs [95,97] was not associated with cognitive function. Three of three studies (3 publications [64,98,99]) found that the PDI and hPDI significantly decreased the risk of cognitive impairment, while two of two studies (2 publications [98,99]) found that the uPDI significantly elevated the risk of cognitive impairment.

In 2 (2 publications [94,96]) of 3 studies (3 publications [26,94,96]), the hPDI significantly decreased the risk of dementia, while the uPDI increased the risk of dementia in 1 study [96]. Higher adherence of the PDI was not associated with dementia risk [96].

de Crom et al. (2023) [94] found no association between the PDI and the risk of dementia when analyzing 9543 individuals with a mean age of 64 years during a follow-up period of 14.5 years, but lower dementia risk with the hPDI was observed only in men. The UK biobank study included 180,532 individuals with a mean age of 57 years during a follow-up period of 10 years. Wu et al. (2023) [96] observed no association between the PDI and the risk of dementia, while a lower risk of dementia with the hPDI and a higher risk of dementia with the uPDI were observed.

Zhang et al. (2023) [26] showed no association between the hPDI and the risk of dementia during a mean follow-up period of 9.4 years when analyzing 114,684 individuals with a mean age of 56 years from the UK Biobank prospective cohort study.

#### 3.3.6. Another Healthy Dietary Pattern

Table 8 summarizes the associations between other healthy dietary patterns and cognitive outcomes in prospective studies. A total of 36 prospective studies (33 publications [16,29,37,41,52,57,70,72,82,92,100,101,102,103,104,105,106,107,108,109,110,111,112,113,114,115,116,117,118,119,120,121,122]) examined the effects of healthy dietary patterns on cognitive outcomes (Table 8).

The cohort studies included in this review are as follows: Rotterdam study [29], Doetinchem Cohort study [57], MAS [110], Wellbeing Eating and Exercise for a Long Life (WELL) study [111], National Institute for Longevity Sciences-Longitudinal Study of Aging (NILS-LSA) project [112], Ohsaki Cohort Study [113], Ohsaki Cohort 2006 Study [109,113], SNAC-K [41,103,106], LRGS-TUA [107], Australian Diabetes, Obesity and Lifestyle (AusDiab) study [115], PIVUS [82], PATH study [16], Osteoporotic Fractures in Men (MrOS) [100], Geisinger Rural Aging Study (GRAS) [116], UK Biobank study [117], CHNS [118,119], MDCS [52], Gothenburg H70 birth cohort study [101], NuAge study [102,105], WII [92], Uppsala longitudinal study [70], AIBL study [72], Taiwan Longitudinal Study of Aging (TLSA) study [104], Hisayama study [121], History-Based Artificial Intelligent Clinical Dementia Diagnostic System (HAICDDS) Project [108], Tzu Chi Vegetarian Study (TCVS) [122], Adventist Health Study-2 (AHS-2) cohort [120], SUN cohort study [37], and NA [114]. The sample size of the 57 studies was 220,234 subjects (range: 132 to 104,895) aged 50.2 to 84 ± 3.7 years, with follow-up periods ranging from 2.33 to 25 years.

In 36 studies (33 publications [16,29,37,41,52,57,70,72,82,92,100,101,102,103,104,105,106,107,108,109,110,111,112,113,114,115,116,117,118,119,120,121,122]), the quality assessment results showed a mean quality assessment score of 7.5 (high quality), with 28 studies (25 publications [16,29,37,41,52,57,70,82,92,101,103,104,106,107,108,109,110,112,113,116,117,118,119,121,122]) scoring high on quality and 8 studies (8 publications [72,100,102,105,111,114,115,120]) scoring moderate.

In total, 9 (8 publications [41,57,92,103,105,111,118,119]) of 20 studies (18 publications [37,41,57,72,82,92,100,102,103,104,105,110,111,114,117,118,119,120]) found that healthy dietary patterns were associated with a higher cognitive function, while no association was found between healthy dietary patterns and the risk of cognitive impairment in 4 (3 publications [16,70,107]) of 5 studies (4 publications [16,70,107,115]).

In terms of dementia, adult subjects with higher adherence to healthy dietary patterns had a lower risk of dementia in 7 (7 publications [101,107,109,112,113,121,122]) of 15 studies (14 publications [29,52,70,92,101,106,107,108,109,112,113,116,121,122]).

The LRGS-TUA study from Malaysia (*n* = 280) indicated that the tropical fruits-oats dietary pattern, which includes high intake of oats along with various tropical fruits, such as orange, banana, papaya, rambutan, and duku, did not show significant association with MCI, but was observed to have an association with decreased risk of dementia incidence [odds ratio (OR) for T3 vs. T1 = 0.101; 95% CI = 0.011, 0.967] [107].

Glans et al. (2023) [52] investigated the association between the Swedish dietary guidelines score (SDGS), which emphasizes dietary fiber, vegetable, fruit, and fish intake, and cautions against the intake of added sugars, red meat, and processed meat, and the development of dementia. They found no significant association between the SDGS and the incidence of all-cause dementia, Alzheimer’s disease dementia, or vascular dementia [52].

Fan et al. (2023) [108] analyzed the association between a vegetarian diet and incident dementia among the Taiwanese population (*n* = 1285). Unexpectedly, a vegetarian diet was associated with a high incidence of all-cause dementia (HR = 1.95; 95% CI = 1.12, 4.30). In particular, vegetarian diets showed a stronger association with vascular dementia (HR = 3.15; 95% CI = 1.10, 9.00) [108].

#### 3.3.7. Western Dietary Pattern (WDP)

Table 9 summarizes the associations between the WDP and cognitive outcomes in prospective studies. In total, 12 studies (12 publications [51,72,92,100,101,102,103,104,105,123,124,125]) observed the effects of the WDP on cognitive outcomes (Table 9).

The cohort studies included in this review are as follows: MrOS [100], SUN cohort study [51], Gothenburg H70 birth cohort study [101], NHS [123], NuAge study [102,105,124], WII [92], ARIC study [125], SNAC-K [103], AIBL study [72], and TLSA study [104]. The sample size of the 36 studies was 51,973 subjects (range: 350 to 16,058), with the mean age ranging from 50.2 ± 6.1 to 74.16 ± 4.16 years. The follow-up duration ranged from 3 to 20 years.

In 12 studies (12 publications [51,72,92,100,101,102,103,104,105,123,124,125]), the quality assessment results showed a mean quality assessment score of 6.9 (moderate quality), with 7 studies (7 publications [51,92,101,103,104,123,125]) scoring high on quality and 5 studies (5 publications [72,100,102,105,124]) scoring moderate.

In total, 7 (7 publications [51,100,102,103,104,105,123]) of 11 studies (11 publications [51,72,92,100,102,103,104,105,123,124,125]) found that higher adherence to the WDP significantly lowered cognitive function.

Rogers-Soeder et al. (2024) [100] (*n* = 4231) provided the results through the Modified Mini-Mental State Examination (3MS) score and Trail B test time in the MrOS cohort. Males in the second quartile had an increased risk of cognitive decline compared with those in the first quartile of the WDP, showing a lower 3MS score [100]. Meanwhile, no significant associations between the WDP and the risk of cognitive decline by Trail B test time were observed [100].

One (1 publication [101]) of two studies (2 publications [92,101]) showed that the WDP increased the risk of dementia, while the other study [92] found no association.

The WII cohort study, which began between 1985 and 1988, assessed dietary intake during 1991–1993, 1997–1999, and 2002–2004 and tracked the development of dementia. This study did not find a significant association between the WDPs and cognitive decline or incidence of dementia [92].

The ARIC study targeting four US communities (Jackson, Mississippi; Forsyth County, North Carolina; suburban Minneapolis, Minnesota; and Washington County, Maryland) investigated the association between midlife dietary patterns and cognitive change for 20 years [125]. The WDP was observed to be associated with lower cognitive functions in the crude model, but the association disappeared after adjusting for health behaviors such as smoking and drinking [125].

#### 3.3.8. Other Dietary Patterns

Table 10 summarizes the associations between other dietary patterns and cognitive outcomes in prospective studies. A total of 16 prospective studies (13 publications [55,82,104,109,114,115,126,127,128,129,130,131,132]) investigated the effects of various dietary patterns on cognitive outcomes (Table 10).

The cohort studies included in this review are as follows: CLHLS [126], Ohsaki Cohort 2006 Study [109], PIVUS [82], UK Biobank study [127,128], CHNS [129,130], Lothian Birth Cohort 1936 study [55], TLSA study [104], WII [132], AusDiab study [115], and NA [114,131]. The sample size of the 36 studies was 776,141 subjects (range: 194 to 497,533), with the mean age ranging from 45 to 86.35 ± 10.20 years. The follow-up duration ranged from 1 to 14.8 years.

In 16 studies (13 publications [55,82,104,109,114,115,126,127,128,129,130,131,132]), the quality assessment results showed a mean quality assessment score of 7.1 (high quality), with 11 studies (11 publications [55,82,104,109,115,126,127,128,129,130,132]) scoring high on quality and 5 studies (2 publications [114,131]) scoring moderate.

In total, 2 (2 publications [109,114]) of 4 studies (4 publications [82,109,114,126]) found neutral effects of the animal-based dietary pattern on cognitive function [114] and the risk of dementia [109].

Tomata et al. (2016) [109] examined the association between the high-dairy pattern and the incidence of dementia in the Japanese elderly population (*n* = 14,402), but no significant association was observed after 4.9 years of follow-up.

A community-based study in Sweden looked at the association between the MED diet and its food groups and cognitive ability [82]. Compared to people with a low intake of meat and meat products, those with a high intake showed poorer performance on the seven-minute screening (7MS) test [82].

Hu et al. (2023) [126] examined the effect of the animal-based diet on the association between green space exposure and cognitive function among the Chinese population, and the study showed the association between animal-based diet scores and cognition using the Mini-Mental State Examination (MMSE). The animal-based diet high in eggs, fish, and meat was associated with an increased risk of cognitive impairment (HR for T3 vs. T1 = 1.64; 95% CI = 1.38, 1.96), and dose response association was also found indicating an 8% increase in the risk of cognitive impairment per each 1-point increase in the animal-based diet score (HR = 1.08; 95% CI = 1.06, 1.09) [126].

Ozawa et al. (2017) [132] analyzed the association between the inflammatory dietary patterns characterized by high consumption of red and processed meat, peas, legumes, and fried food, and low consumption of whole grains, and cognitive function in 5083 subjects from the WII cohort study. In reasoning and global cognition, people in the highest tertile of the inflammatory diet pattern score showed more rapid cognitive decline than those in the lowest tertile [132].

## 4. Discussion

The aim of this review was to investigate the effects of different diary patterns on cognitive function, cognitive impairment, and dementia, as well as to provide updated and integrated findings by extensively examining prospective cohort studies.

The MED diet improved cognitive function, but did not favorably influence cognitive impairment and dementia. Moreover, the DASH diet reduced cognitive impairment, but did not beneficially affect cognitive function and dementia. Fekete et al. (2025) [133] conducted a meta-analysis of cohort, case-control, and cross-sectional studies. In line with our findings, they also found that higher adherence to the MED diet was associated with a delay in cognitive decline (HR = 0.82; 95% CI = 0.75, 0.89) in a meta-analysis of 13 observational studies [43,66,69,70,73,79,87,134,135,136,137,138,139], and it decreased the risk of incident dementia by 11% (HR = 0.89; 95% CI = 0.83, 0.95) in a meta-analysis of 10 observational studies [29,38,44,47,52,54,69,70,86,140].

Comparing the highest with the lowest categories, the present review found that the MIND diet significantly improved cognitive function. Moreover, the MIND diet significantly lowered cognitive impairment. This finding was observed in a meta-analysis of [33,34,35,37,40,42,141] conducted by Huang et al. (2023) [28]. This meta-analysis included 26,103 subjects aged 45 years from 8 prospective cohort studies across 3 countries of the USA (*n* = 6), Spain (*n* = 1), and China (*n* = 1). The MIND score was associated with improved cognitive function (β = 0.042, 95% CI = 0.020, 0.065; I^2^ = 39.5%, P_heterogeneity_ = 0.142) [28].

Even though we observed improvement in cognitive function and cognitive impairment, which are risk factors of dementia, we could not observe beneficial effects of the MIND diet on dementia. Inconsistent with our findings, Chen et al. (2023) [27] observed the association between the MIND diet score and decreased risk of dementia when comparing the highest tertile with the lowest tertile (pooled HR = 0.83; 95% CI = 0.76, 0.90; I^2^ = 35%) in a meta-analysis of 11 cohort studies reported in 4 publications [29,31,32,142] with 224,049 participants (5279 incident dementia cases). The discrepancy in the findings of our review and of Chen et al. (2023) [27] could be explained by the fact that we included many more prospective studies. Moreover, the prospective studies included in this review had differences in dietary assessment methods, the scoring of the MIND components, study design, and so on.

The MIND diet is a dietary pattern in which the cardiovascular protective MED and DASH diets are combined for brain health. Morris et al. (2015) [42,143] established the total MIND diet score by summing scores of 0, 0.5, or 1 over the dietary component of the MED and DASH diets. The MIND diet emphasizes healthy food components and limits unhealthy food components. Healthy food components consist of whole grains, beans, seafood, non-fried poultry, non-fried fish, green leafy vegetables, other vegetables, wine, nuts, berries, and olive oil. Unhealthy food components include red meat, butter and stick margarine, cheese, pastries and sweets, and fried/fast food (French fries, chicken nuggets) [42,143].

We clearly observed that the MIND, MED, and DASH diets could improve cognitive function and cognitive impairment, which indicates that these diets, especially the MIND diet, exert a protective role in brain aging.

The mechanisms underlying the protective effects of the MIND diet on brain health point to food components rich in antioxidants and anti-inflammatory nutrients and nutrients. Vitamins A, E, C, and minerals rich in the MIND diet can exert a protective role from oxidative stress in the brain [144]. An observational study has shown that green leafy vegetables abundant in vitamin K, folate, vitamin E, lutein, nitrate, polyphenols, and nutrients can delay brain aging [145]. Moreover, a clinical trial showed that the supplementation of folate and vitamin B12 deceased cognitive impairment and inflammation in subjects with Alzheimer’s disease (AD) [146]. Animal [147] and human [148] studies have shown that intake of berries can delay cognitive decline. Animal studies have indicated that vitamin E might play a role in healthy cognitive function by inhibiting lipid peroxidation [149], oxidative stress [150,151], neuron loss [152], beta-amyloid accumulation [153]. Human studies have also indicated that vitamin E supplements could enhance cognitive function [154]. Healthy food components from fish, nuts, and olive oil are rich in polyunsaturated fatty acids (PUFAs), especially omega-3 PUFAs [e.g., docosahexaenoic acid (DHA)], which can reduce cognitive impairment and the risk of dementia [155,156,157,158] in humans. Their effects can be elucidated through protection from oxidative stress and inflammation, neurotransmission modulation, enhancement of neurogenesis, and neuronal survival [159,160].

The MIND diet, emphasizing whole grains, fruits, vegetables, and legumes, induces increased consumption of high-quality carbohydrates and dietary fiber, and can eventually delay cognitive decline [161,162,163,164,165]. We observed an association between the hPDI and reduction in cognitive impairment and dementia, as well as an association between the uPDI and an increase in cognitive impairment. The MIND diet encourages adherence to plant-based dietary patterns, composed of fruits, vegetables, legumes, nuts, and whole grains. These dietary patterns are abundant in antioxidants, vitamins, polyphenols, other phytochemicals, and unsaturated fatty acids [166,167,168,169,170,171].

A possible explanation for why we observed no beneficial association between the MIND diet and incident dementia can be suggested. The association between the MIND diet and incident dementia could be an interplay among various factors, as shown in multidomain RCTs [172,173]. Recent RCTs [172,173,174] emphasize the importance of intensive lifestyle modification in order to improve dementia risk factors, including cognitive decline and cognitive impairment, leading to a reduction in dementia risk.

Baker et al. (2025) [172] conducted a single-blind, multicenter RCT (the US POINTER study) involving 2111 older subjects aged 60–79 years with normal memory and thinking but at risk of cognitive decline and dementia. The study was conducted at 5 clinical sites in the USA. In this RCT, the subjects were divided into a structured group and a self-guided structured group (*n* = 1056). The structured group participated in an intensive program encouraging aerobics (4 days per week, 30–35 min per session), adherence to the MIND diet, blueberry intake, online cognitive training, mandatory social engagement, and result review of blood pressure and hemoglobin A1c. On the other hand, the self-guided group (*n* = 1055) was encouraged to come up with physical and cognitive activity, a healthy diet, social engagement, and cardiovascular health monitoring in their own way. The structured group showed significantly improved global cognition over 2 years compared with the self-guided group, which indicated a holistic impact of physical activity, the MIND diet, and social interaction on cognitive improvement in normal brain aging [172].

Ornish et al. (2024) [173] conducted the first RCT to determine if intensive lifestyle intervention beneficially influences cognitive function in 51 subjects (mean age 73.5 years) with MCI or early dementia due to AD. The intervention group was involved in walking and mild strength exercises, stress management classes, supplement intake of omega-3 fatty acids with curcumin, multivitamin and minerals, coenzyme Q10, vitamin C, vitamin B12, magnesium L-threonate, hericium erinaceus, probiotics, and a whole food plant-based (vegan) diet. A whole food plant-based (vegan) diet is rich in fruits, vegetables, whole grains, legumes, soy products, seeds, and nuts, and low in saturated fats, sweeteners, and refined grains. It draws 63–68% of its calories from complex carbohydrates, 14–18% from fats, and 16–18% from protein. In the findings, the cognitive function and plasma Aβ42/40 ratio increased in the intervention group, whereas the cognitive function and plasma Aβ42/40 ratio decreased in the control group. The increased cognitive function and plasma Aβ42/40 ratio in the intervention group were correlated with desired lifestyle changes at 20 weeks. The microbiome configuration was beneficially changed only in the intervention group after 20 weeks. This RCT indicated that intensive and comprehensive lifestyle changes could enhance cognitive function in the elderly with MCI or early dementia due to AD [173].

The present review found that higher adherence to the WDP significantly deteriorated cognitive outcomes when comparing the highest categories with the lowest, which indicates that the WDP exerts a negative role in brain aging. The mechanisms underlying the negative effects of the WDP on brain health include increased risks of obesity, cardiometabolic disease, oxidative stress, systemic inflammation, gut microbiota dysbiosis, blood–brain barrier dysfunction, neuroinflammation, and amyloid accumulation [175,176,177], which could be attributable to the components of the WDP, which is high in refined grains, red and processed meat, high-fat dairy, sugary beverages, and sweets. In particular, the sodium, saturated fatty acids, advanced glycation end products (AGEs), and trimethylamine N-oxide (TMAO) derived from red and processed meat could be mechanistic links to deteriorated cognitive function [178,179,180,181,182].

Two UK biobank studies included in the present review addressed unhealthy dietary patterns [127,128]. Xu et al. (2023) [128] showed no association between poor dietary patterns and the risk of dementia during a mean follow-up period of 14.8 years. They defined the poor diet pattern as the consumption of less than 4 of 7 dietary components of refined grain, whole grain, fish, unprocessed meat, processed meat, fruits, and vegetables.

Meanwhile, Zhang et al. (2024) [127] observed a significant association between the high-sugar dietary score and the risk of all-cause dementia during a mean follow-up period of 11.8 years. The high-sugar dietary score was calculated by identifying the high-sugar dietary pattern high in fresh fruit, sugar-sweetened beverages, and other sugary drinks, fruit juice, dried and stewed fruit, table sugars and preserves, milk-based and powdered drinks, chocolate, and confectionery. The possible mechanisms linking the high-sugar dietary pattern to the risk of dementia can be proposed. A high-sugar diet can elevate brain insulin resistance, which deteriorates brain function by inducing glial cell dysfunction, neuroinflammation, and beta-amyloid plaque accumulation [183,184,185]. Moreover, a study conducted in vitro and in vivo showed that a high-sugar diet induced gut microbiota dysbiosis [186], leading to the incidence of dementia with the interactive pathways of oxidative stress, metabolic dysfunction, and neuroinflammation through the microbiota–gut–brain axis [187].

This review encompasses the latest research on the relationship between various dietary patterns and cognitive health, including prospective cohort studies published up to October 2024 on the association between dietary patterns and cognitive function, cognitive impairment, and dementia. Most of the included studies were of good quality, scoring seven or higher on the quality assessment, and a significant number of studies reported changes in cognitive function over a follow-up period of 5 years or more. This review may contribute to a broader understanding of the commonalities and differences in the impact of various dietary patterns on cognitive health.

Several limitations should be considered when interpreting this review’s results. Although this study attempted to minimize bias by including only prospective cohort studies, the nature of observational studies leaves room for residual confounding factors. When presented with multiple outcome values in the original study, we included the most heavily adjusted values in our review to minimize the influence of confounding factors. However, each study reported cognitive health through various types of testing methods, resulting in heterogeneity in the presentation of results. This review tried to organize and understand the results by dividing them into cognitive decline, cognitive impairment, and the incidence of dementia. However, there might be a lack of focus on the results for each detailed cognitive domain. Lastly, this review was limited to only publications in which the full text was available in English.

## 5. Conclusions

This review provides updated and integrated prospective cohort evidence for the beneficial effect of the MIND diet combined with the MED and DASH diets on cognitive function and cognitive impairment. Moreover, the hPDI is associated with reductions in cognitive impairment and dementia. The uPDI is associated with an increased risk of cognitive impairment. The WDP is associated with an increased risk of cognitive function. Further prospective cohort studies should be conducted considering healthy and unhealthy dietary patterns to establish definitive evidence for the association between various diets and cognitive health.

## Figures and Tables

**Figure 1 nutrients-17-03469-f001:**
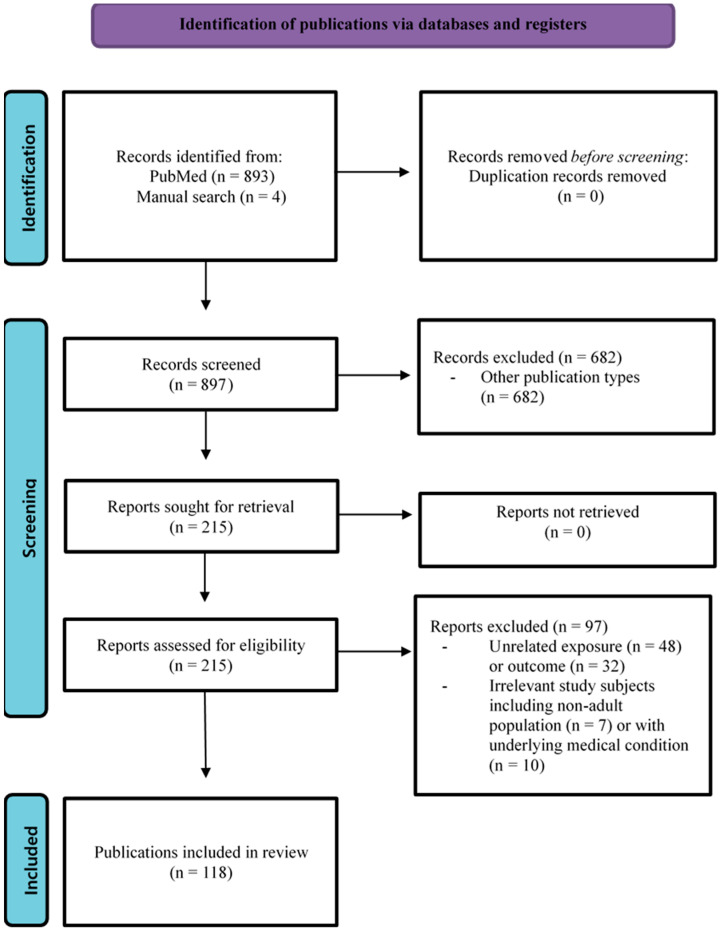
The flow diagram of screening and selection for this review.

**Table 1 nutrients-17-03469-t001:** Search strategy for this review.

Exposure (#1)		Outcome (#2)	#3
MIND	AND	cognition	#1 AND #2
DASH		cognitive function	
HEI		cognitive decline	
AHEI		cognitive impairment	
Mediterranean diet		mild cognitive impairment	
Vegetarian diet		dementia	
Ketogenic diet			
Plant-based diet			
Animal-based diet			
Dairy-based diet			
Processed meat diet			
Fruit and vegetable diet			
Western diet			

AHEI, alternative healthy eating index; DASH, Dietary Approaches to Stop Hypertension; HEI, healthy eating index; MIND, Mediterranean-Dietary Approaches to Stop Hypertension Intervention for Neurodegenerative Delay.

**Table 2 nutrients-17-03469-t002:** Inclusion and exclusion criteria for study selection in this review.

Category	Inclusion Criteria	Exclusion Criteria
Date of publication	Studies published up to 30 October 2024	Studies published after 30 October 2024
Language	Full-text available in English	Full-text available in non-English
Publication type	Cohort studies Follow-up studies Longitudinal studies Observational prospective studies	Case-controls Case reports Commentaries Cross-sectional studies Editorials Interventions Meta-analysis RCTs Reviews
Study subjects	Human subjects	Non-human subjects (in vivo or in vitro)
Age of study subjects	Age at exposure: adults (aged 20 years and older)	Age at exposure: infants to adolescents (birth to 19 years)
Exposure	Studies that measure the consumption of and/or adherence to a dietary pattern (indices/scores)	Studies that examine the consumption of food by food groups Studies that examine the effects of medication
Outcomes	Cognitive function change - Global cognition score - MMSE - STICS-m score - Other tests that examine the cognitive function of subjects Cognitive impairment incidence - Cognitive impairment or MCI Dementia incidence - All-cause dementia	Alzheimer’s disease Brain volume
Health status of study subjects	Studies that recruit subjects who are healthy and/or not diagnosed with dementia at baseline	Studies that recruit subjects who are diagnosed with a certain disease

MCI, mild cognitive impairment; MMSE, Mini-Mental State Examination; RCTs, randomized controlled trials; STICS, Spanish Telephone Interview for Cognitive Status.

**Table 3 nutrients-17-03469-t003:** Summary of the 28 publications (36 studies) prospective studies that investigated associations between the MIND diet and cognitive outcomes.

Author, Year, Region	Study Name	Adherence	Subjects			Study Period (Follow-Up Years)	Outcomes			Study Quality
			Total (*n*)	Female (%)	Age (Range or Mean/SD or Median) (Years)	Average Follow-Up (Year)	Cognitive Function	Cognitive Impairment or MCI	Dementia	
Li et al., 2024, USA [15]	ROS and MAP	MIND diet score	1204 All participants with RNA-Seq data	68.0	80.8 ± 6.9	8.8		↔MCI in the fully adjusted model OR (95% CI; *p* value) 0.94 (0.81, 1.10; *p* = 0.48)	↓Dementia risk in the fully adjusted model OR (95% CI; *p* value) 0.77 (0.67, 0.88; *p* = 0.0002)	7
			444 Subset of participants with dietary and RNA-Seq data	70.5	82.5 ± 6.0	9.1		↓MCI in the fully adjusted model OR (95% CI; *p* value) 0.76 (0.59, 0.91; *p* = 0.003)	↓Dementia risk in the fully adjusted model OR (95% CI; *p* value) 0.66 (0.52, 0.84; *p* = 0.0009)	7
			722 Independent set of participants with RNA-Seq data	66.3	79.7 ± 7.2	8.3		↔MCI in the fully adjusted model OR (95% CI; *p* value) 0.89 (0.72, 1.11; *p* = 0.3).	↓Dementia risk in the fully adjusted model OR (95% CI; *p* value) 0.76 (0.59, 0.97; *p* = 0.03)	7
O’Reilly et al., 2024, Australia [16]	PATH study	MIND diet score	1753	Low: 45 Medium: 53 High: 57	60–64	12		↓MCI in the fully adjusted model comparing highest vs. lowest intake OR (95% CI) T1 1 (Ref) T2 0.65 (0.41, 1.03) T3 0.60 (0.37, 0.99)		7
Sawyer et al., 2024, USA [17]	REGARDS	MIND diet score	14,145	56.7	64.0 ± 9.0	10.92		↓Cognitive impairment in the fully adjusted model comparing highest vs. lowest intake OR (95% CI; *p* value) T1 (Ref) T2 0.93 (0.82, 1.06; *p* = 0.91) T3 0.85 (0.74, 0.99; *p* = 0.06)		8
Agarwal et al., 2024, USA [18]	CHAP	MIND diet score	5259	62	73.5 ± 6.0	7.8	↑Cognitive function in the fully adjusted model comparing highest vs. lowest intake β (95% CI) T1 (Ref) T2 0.0044 (−0.002, 0.012) T3 0.0083 (0.002, 0.015)			7
Seago et al., 2024, USA [19]	HRS	MIND diet score	5143	60	69 ± 10	7	↔Cognitive function in the fully adjusted model β (95% CI; *p* value) 0.02 (0, 0.04; *p* = 0.094)			6
Bhave et al., 2024, USA [20]	REGARDS	MIND diet score	14,175	Non-cases: 57.9 Cases: 59.6	Non-cases: 63.4 ± 8.6 Cases: 65.8 ± 8.8	Non-cases: 10.9 Cases: 7.5		↓Cognitive impairment in the fully adjusted model HR (95% CI; *p* value) 0.91 (0.87, 0.95; *p* < 0.00001)		8
Thomas et al., 2024, USA [21]	FHS Offspring cohort	MIND diet score	1644	54	69.6 ± 6.9	14			↓Dementia incidence for each 1-SD increase in MIND diet score per 10,000 person-years of follow-up SD (95% CI) −33.6 (−55.6, −11.7)	6
M. Zapawi et al., 2024, Malaysia [22]	LRGS-TUA and FRGS	MY-MINDD scores	810		67.9 ± 4.7	NA		↓MCI in the fully adjusted model OR (95% CI) Q1 1 (Ref) Q2 0.52 (0.33, 0.84) Q3 0.50 (0.33, 0.77) Q4 0.43 (0.26, 0.72)		6
Sager et al., 2024, European countries (Switzerland, Germany, Austria, France, and Portugal) [23]	DO-HEALTH clinical trial	MIND diet score	2028	60.5	74.88 ± 4.42	3		↔MCI in the fully adjusted model MoCA< 26 OR (95% CI; *p* value) 0.99 (0.94, 1.04; *p* = 0.62) MoCA < 24 1.03 (0.96, 1.1; *p* = 0.426)		6
Lin et al., 2024, China [24]	CLHLS	cMIND diet score	6411	51.0	80.61 ± 10.0	3		↓Cognitive impairment in fully adjusted model comparing highest vs. lowest intake OR (95% CI) Q1 1 (Ref) Q2 0.94 (0.76, 1.17) Q3 0.87 (0.71, 1.07) Q4 0.77 (0.60, 0.97)		7
McEvoy et al., 2024, UK and Ireland [25]	TwinsUK	MIND diet score	220	100	51.9 ± 12.5	10	↑Cognitive function per 1-point increase in MIND diet score in the fully adjusted model PAL β (95% CI; *p* value) −1.75 (−2.96, −0.54; *p* = 0.005)			6
Zhang et al., 2023, UK [26]	UK Biobank Study	MIND diet score	114,684	55.5	56.8 ± 7.77	9.4			↔Dementia incidence in the fully adjusted model comparing highest vs. lowest intake HR (95% CI; *p* value) T1 (Ref) T2 0.91 (0.73, 1.14; *p* = 0.4) T3 0.89 (0.71, 1.12; *p* = 0.3)	9
Chen et al., 2023, USA [27]	WII	MIND diet score	8358	30.9	62.2 ± 6.0	12.9			↔Dementia incidence in the fully adjusted model comparing highest vs. lowest intake HR (95% CI) T1 (Ref) T2 1.03 (0.73, 1.45) T3 0.96 (0.66, 1.38)	7
	HRS		6758	58.7	66.5 ± 10.4	5.0			↔Dementia incidence in the fully adjusted model comparing highest vs. lowest intake HR (95% CI) T1 (Ref) T2 0.95 (0.73, 1.25) T3 0.83 (0.63, 1.09)	8
	FHS Offspring cohort		3020	54.6	64.2 ± 9.1	10.7			↔Dementia incidence in the fully adjusted model comparing highest vs. lowest intake HR (95% CI) T1 (Ref) T2 0.96 (0.70, 1.33) T3 0.69 (0.48, 0.99)	8
Huang et al., 2023, China [28]	CHNS	MIND diet score	4066	50.5	62.2	3	↑Cognitive function in the fully adjusted model comparing highest vs. lowest intake β (95% CI) 0.010 (0.000, 0.020)			8
de Crom et al., 2022, The Netherlands [29]	Rotterdam Study	MIND diet score	Baseline I: 5375	Baseline I: 59.0	Baseline I: 67.7 ± 7.8	Baseline I: 15.6			↔Dementia risk in Baseline I in the fully adjusted model comparing highest vs. lowest intake HR (95% CI) 0.99 (0.94, 1.05)	9
			Baseline II: 2861	Baseline II: 57.4	Baseline II: 75.3 ± 5.9	Baseline II: 5.9			↓Dementia risk in Baseline II in the fully adjusted model comparing highest vs. lowest intake HR (95% CI) 0.79 (0.70, 0.91)	9
Cornelis et al., 2022, UK [30]	UK Biobank Study	MIND diet score	120,661	56.5	T1: 57.3 ± 8.0 T2: 57.9 ± 7.9 T3: 58.3 ± 7.7	10.5	↑Cognitive function in the fully adjusted model comparing highest vs. lowest intake -FI test β (95% CI; *p* value) T1 (Ref) T2 −0.03 (−0.07, 0.007; *p* = 0.12) T3 −0.14 (−0.18, −0.10; *p* < 0.0001) -Pairs matching test T1 (Ref) T2 0.01 (0.001, 0.02; *p* = 0.03) T3 0.03 (0.02, 0.04; *p* < 0.0001) -SDS test T1 (Ref) T2 −0.07 (−0.15, 0.02; *p* = 0.16) T3 −0.25 (−0.33, −0.16; *p* < 0.0001) -Trail A test T1 (Ref) T2 0.005 (−0.001, 0.01; *p* = 0.0002) T3 0.01 (0.007,0.02; *p* < 0.0001) -Trail B test T1 (Ref) T2 0.01 (0.005,0.02; *p* = 0.0002) T3 0.02 (0.02,0.03; *p* < 0.0001)		↔Dementia Incidence in the fully adjusted model comparing highest vs. lowest intake HR (95%CI; *p* value) T1 (Ref) T2 1.06 (0.90,1.24; *p* = 0.51) T3 0.90 (0.74,1.09; *p* = 0.27)	9
Vu et al., 2022, USA [31]	CHAP-white	MIND diet score	2449	T1: 52 T2: 65 T3: 67	T1: 74.0 ± 6.3 T2: 74.2 ± 6.3 T3: 72.2 ± 5.7	20	↔Cognitive function in the fully adjusted model comparing highest vs. lowest intake β (95% CI; *p* value) T1 (Ref) T2 0.0001 (−0.01, 0.01; *p* = 0.99) T3 −0.0008 (−0.01, 0.01; *p* = 0.89)		↔Dementia incidence in the fully adjusted model comparing highest vs. lowest intake OR (95% CI) T1 (Ref) T2 0.87 (0.30, 2.54) T3 1.23 (0.47, 3.18)	8
	CHAP-black		2449	T1: 54 T2: 66 T3: 69	T1: 71.7 ± 4.6 T2: 71.9 ± 4.5 T3: 71.1 ± 4.1	20	↔Cognitive function in the fully adjusted model comparing highest vs. lowest intake β (95% CI; *p* value) T1 (Ref) T2 0.0003 (−0.01, 0.01; *p* = 0.95) T3 −0.003 (−0.01, 0.01; *p* = 0.51)		↔Dementia incidence in the fully adjusted model comparing highest vs. lowest intake OR (95% CI) T1 (Ref) T2 0.86 (0.36, 2.05) T3 1.48 (0.51, 4.27)	8
	MAP		725	T1: 73 T2: 74 T3: 77	T1: 82.3 ± 7.2 T2: 82.5 ± 6.5 T3: 80.3 ± 6.8	20	↑Cognitive function in the fully adjusted model comparing highest vs. lowest intake β (95% CI; *p* value) T1 (Ref) T2 0.006 (−0.01, 0.02; *p* = 0.5) T3 0.03 (0.01, 0.05; *p* = 0.001)		↓Dementia incidence in the fully adjusted model comparing highest vs. lowest intake HR (95% CI; *p* value) T1 (Ref) T2 0.85 (0.62, 1.16; *p* = 0.31) T3 0.63 (0.42, 0.92; 0.02)	7
	WHIMS		5308	100	T1: 69.8 ± 3.8 T2: 70.2 ± 3.85 T3: 70.3 ± 3.8	20			↓Dementia incidence in the fully adjusted model comparing highest vs. lowest intake HR (95% CI; *p* value) T1 (Ref) T2 0.87 (0.79, 0.97; *p* = 0.008) T3 0.80 (0.72, 0.89; *p* < 0.0001)	7
Thomas et al., 2022, France [32]	3C Bordeaux study	French-adapted MIND diet score	1412	63.0	75.8 ± 4.8	9.7			↓Dementia incidence in the fully adjusted model comparing highest vs. lowest intake HR (95% CI) T1 (Ref) T2 0.93 (0.73, 1.17) T3 0.73 (0.55, 0.97) HR for 1-point score (95% CI) 0.90 (0.83, 0.96)	8
Boumenna et al., 2022, USA [33]	BPRHS	MIND diet score	573	70	57.2 ± 7.9	8	↑Cognitive function in the fully adjusted model comparing highest vs. lowest intake β (95% CI) Q1 (Ref) Q2 0.005 (−0.053, 0.064) Q3 0.006 (−0.043, 0.055) Q4 0.047 (−0.006, 0.099) Q5 0.093 (0.035, 0.152) *p* trend = 0.0019			8
Dhana et al., 2021, USA [34]	MAP	MIND diet score	569	70.5	age at death: 90.8 ± 6.1		↑Global cognition proximate to death in higher MIND diet score β (SE; *p* value) 0.119 (0.040; *p* = 0.003)			5
Melo van Lent et al., 2021, USA [35]	FHS	MIND diet score	1584	54	61 ± 9	6.6 ± 1.1	↔Global Cognition β (SE; *p* value) −0.002 (0.02; *p* = 0.87)			8
Nishi et al., 2021, Spain [36]	PREDIMED-Plus trial	MIND diet score	4674	48	65	2	↑Cognitive function for DST-B β (95% CI; *p* trend) 0.058 (0.002, 0.114; *p* trend = 0.045) ↔MMSE, GCF, CDT, VFT-a, VFT-p, TMT-a, TMT-b, DST-f			7
Munoz-Garcia et al., 2020, Spain [37]	SUN cohort study	MIND diet score	806	34	67	6	↔Cognitive function for STICS-m score change in the fully adjusted model comparing highest vs. lowest intake β (95% CI) T1 0 (Ref) T2 0.17 (−0.28, 0.62) T3 0.47 (−0.07, 1.02) ↑Cognitive function for each 1.5 points (0–15) in the fully adjusted model β (95% CI; *p* value) 0.27 (0.05, 0.48; *p* < 0.05)			7
Hosking et al., 2019, Australia [38]	PATH study	MIND diet score	1220	T1: 42 T2: 51 T3: 60	T1: 62.4 ± 1.5 T2: 62.5 ± 1.5 T3: 62.5 ± 1.5	12		↓MCI in the fully adjusted model comparing highest vs. lowest intake OR (95% CI) T1 (Ref) T2 0.94 (0.57, 1.56) T3 0.47 (0.24, 0.91)		7
Adjibade et al., 2019, France [39]	NutriNet-Santé study	MIND diet score	6011	60	64.4 ± 4.3	6	↔SMC in the fully adjusted model comparing highest vs. lowest intake HR (95% CI) T1 (Ref) T2 0.97 (0.84, 1.12) T3 0.94 (0.79, 1.11)			8
Berendsen et al., 2018, USA [40]	NHS	MIND diet score	16,058	100	74.3 ± 2.3	6	↑Verbal memory score comparing highest vs. lowest intake MD (95% CI; *p* trend) 0.04 (0.01, 0.07; *p*-trend = 0.02) ↔Global cognitive and/or TICS scores ↔Global cognitive, verbal memory, and/or TICS score in long-term effect			7
Shakersain et al., 2018, Sweden [41]	SNAC-K	MIND diet score	2223	60.8	Men: 69.5 ± 8.6 Women: 71.3 ± 9.1	6	↑MMSE score in the fully adjusted model β (95% CI; *p* value) Moderate intake: 0.075 (0.012, 0.138; *p* = 0.019) High intake: 0.126 (0.064, 0.188; *p* < 0.001)			8
Morris et al., 2015, USA [42]	MAP	MIND diet score	960	75	81.4 ± 7.2	4.7	↑Cognitive function β (SE; *p* value) 0.0092 (0.0022; *p* < 0.0001)			6

3C, Three-City; BPRHS, Boston Puerto Rican Health Study; CDT, clock-drawing test; CHAP, Chicago Health and Aging Project; CHNS, China Health and Nutrition Survey; CI, confidence interval; CLHLS, Chinese Longitudinal Healthy Longevity Surveys; cMIND, Chinese version of the Mediterranean-DASH intervention for neurodegenerative delay; DO-HEALTH, Vitamin D3–Omega-3–Home Exercise–Healthy Ageing and Longevity Trial; DST-B, Digit Span Test—backward; DST-f, Digit Span Test—forward; FHS, Framingham Heart Study; FI, fluid intelligence; FRGS, Fundamental Research Grant Scheme; GCF, global cognitive function; HR, hazard ratio; HRS, Health and Retirement Study; LRGS-TUA, Long-Term Research Grant Scheme—Towards Useful Aging; MAP, Rush Memory and Aging Project; MCI, mild cognitive impairment; MD, mean difference; MIND, Mediterranean-DASH intervention for neurodegenerative delay; MMSE, Mini-Mental State Examination; MoCA, Montreal Cognitive Assessment; MY-MINDD, Malaysian version of the Mediterranean-DASH intervention for the neurodegenerative delay diet; n, number; NA, not available; NHS, Nurses’ Health Study; OR, odds ratio; PAL, paired-associates learning; PATH, Personality and Total Health Through Life Cohort; PREDIMED, PREvención con DIeta MEDiterránea; Q, quintile; Ref, Reference; REGARDS, REasons for Geographic and Racial Differences in Stroke; RNA-Seq, ribonucleic acid sequencing; ROS, Religious Orders Study; SD, standard deviation; SDS, symbol digit substitution; SE, standard error; SMC, subjective memory complaint; SNAC-K, Swedish National study on Aging and Care in Kungsholmen; STICS-m, Spanish Telephone Interview for Cognitive Status; SUN, Seguimiento Universidad de Navarra; T, tertile; TICS, telephone interview of cognitive status; TMT-a, Trail Making Test, part a; TMT-b, Trail Making Test, part b; TwinsUK, United Kingdom Adult Twin Registry; UK, United Kingdom; USA, United States of America; VFT-a, verbal fluency tasks—semantical; VFT-p, verbal fluency tasks—phonological; WHIMS, Women’s Health Initiative Memory Study; WII, Whitehall II study; β, beta coefficient; ↑, significant increase in outcome; ↓, significant decrease in outcome; ↔, no significant effect.

**Table 4 nutrients-17-03469-t004:** Summary of the 55 publications (57 prospective studies) that investigated associations between MED diet and cognitive outcomes.

Author, Year, Region	Study Name	Adherence	Subjects			Study Period (Follow-Up Years)	Outcomes			Study Quality
			Total (*n*)	Female (%)	Age (Range or Mean/SD or Median) (Years)	Average Follow-Up (Year)	Cognitive Function	Cognitive Impairment or MCI	Dementia	
Seago et al., 2024, USA [19]	HRS	MDS	6154	60	69 ± 10	7	↑Cognitive function β (95% CI; *p* value) 0.03 (0.01, 0.05; *p* = 0.002)			6
Bhave et al., 2024, USA [20]	REGARDS	MDS	14,175	Non-cases: 57.9 Cases: 59.6	Non-cases: 63.4 ± 8.6 Cases: 65.8 ± 8.8	Non-cases: 10.9 Cases: 7.5		↔Cognitive impairment in the fully adjusted model		8
McEvoy et al., 2024, UK and Ireland [25]	TwinsUK	MDS	220	100	51.9 ± 12.5	10	↑Cognitive function per 1-point increase in MDS in the fully adjusted model ▪PAL β (95% CI; *p* value) −1.67 (−2.71, −0.65; *p* < 0.01)			6
Feng et al., 2024, China [43]	NA	MDS	1648	49	≥60	3	↑Cognitive function in the fully adjusted model β (SE; *p* value) MMSE −0.020 (0.009; *p* = 0.026)			8
Zhang et al., 2023, UK [26]	UK Biobank Study	MDS	114,684	55.5	56.8 ± 7.77	9.4			↔Dementia incidence in the fully adjusted model comparing highest vs. lowest intake HR (95% CI; *p* value) T1 (Ref) T2 0.99 (0.81, 1.22; *p* = 0.937) T3 0.94 (0.74, 1.19; *p* = 0.609)	9
Shannon et al., 2023, UK [44]	UK Biobank study	MEDAS Score	60,298	48.5	63.8 ± 2.7	9.1			↓Dementia incidence in the fully adjusted model HR (95% CI) T1 (Ref) T2 0.90 (0.79, 1.08) T3 0.77 (0.65, 0.91)	9
		PYRAMID score							↓Dementia incidence in the fully adjusted model HR (95% CI) T1 (Ref) T2 0.99 (0.85, 1.16) T3 0.86 (0.73, 1.02)	
de Crom et al., 2022, The Netherlands [29]	Rotterdam Study	MDS	Baseline I: 5375	Baseline I: 59.0	Baseline I: 67.7 ± 7.8	Baseline I: 15.6			↔Dementia incidence during Baseline I in the fully adjusted model HR (95% CI) 1.04 (0.97, 1.10)	9
			Baseline II: 2861	Baseline II: 57.4	Baseline II: 75.3 ± 5.9	Baseline II: 5.9			↓Dementia incidence during Baseline II in the fully adjusted model HR (95% CI) 0.75 (0.66, 0.86)	9
Vlachos et al., 2022, Greece [45]	HELIAD study	MDS	939	60.8	72.96 ± 4.95	3.1	↑Cognitive function in the fully adjusted model β (MDS × time), *p* value −0.007 (*p* = 0.005)			7
Gregory et al., 2022, Europe [46]	EPAD LCS	MEDAS scores	1826	56.2	65.69 ± 7.42	5	↑Cognitive function in the fully adjusted model ▪ FMT β (95% CI; *p* value) 0.10 (0.04, 0.17; *p* = 0.002)			7
Mamalaki et al., 2022, Greece [47]	HELIAD study	MDS	1018	60	73.1 ± 5.0	3	↑Global cognition score in the fully adjusted model comparing highest vs. lowest intake β (95% CI; *p* value) Q1 (Ref) Q2 −0.010 (−0.040, 0.021; *p* = 0.534) Q3 0.018 (−0.010, 0.047; *p* = 0.208) Q4 0.054 (0.030, 0.078; *p* < 0.001)		↓Dementia incidence RR (95% CI; *p* value) Q1 (Ref) Q2 0.977 (0.961, 0.994; *p* = 0.007) Q3 0.984 (0.967, 1.001; *p* = 0.065) Q4 0.968 (0.955, 0.982; *p* < 0.001)	7
Moustafa et al., 2022, USA [48]	HCHS/SOL study of SOL–INCA	MDS	6321	57.8	56.1 ± 0.18	7	↑Cognitive function in the fully adjusted model comparing highest vs. lowest intake β (95% CI) ▪B-SEVLT Sum 0.12 (0.05, 0.20) ▪B-SEVLT Recall 0.14 (0.05, 0.23) ↔Global cognition score 0.04 (−0.01, 0.09) ↔Word fluency −0.05 (−0.12, 0.02) ↔DSST score −0.01 (−0.06, 0.04)			9
Chen et al., 2022, Australia [49]	MAS	MDS	1037	55.2	78.8 ± 4.8	6	↔Global cognition ↔Cognitive function ↔Specific domain scores			8
Yuan et al., 2022, USA [50]	NHS	aMDS	49,493	100	48 ± 7	31	↑Cognitive function in the fully adjusted model comparing highest vs. lowest intake ▪Moderate SCD OR (95% CI) Q1 1.00 (Ref) Q2 0.97 (0.92, 1.04) Q3 0.94 (0.89, 1.01) Q4 0.93 (0.87, 1.00) Q5 0.81 (0.75, 0.87) ▪Severe SCD Q1 1.00 (Ref) Q2 0.87 (0.79, 0.96) Q3 0.82 (0.74, 0.90) Q4 0.74 (0.67, 0.83) Q5 0.57 (0.51, 0.64)			7
Muñoz-García et al., 2022, Spain [51]	SUN cohort study	MDP	806	30	66 ± 5.5	6	↓Cognitive function for STICS-m score change in the fully adjusted model comparing highest vs. lowest intake ▪STICS-m score β (95% CI; *p* value) T1 0 (Ref) T2 0.16 (−0.34, 0.66) T3 0.71 (0.15, 1.26; *p* = 0.01)			7
Glans et al., 2023, Sweden [52]	MDCS	mMDS: 0–10	28,025	60.7	58.1 ± 7.6	19.8			↔Dementia incidence in the fully adjusted model HR (95% CI) 0.95 (0.76, 1.18)	9
Wade et al., 2021, USA [53]	MSLS	MDS	530	62.8	61.6 ± 11.8	5	↑GCF in the fully adjusted model (≥ 70 years) β (*p* value) −0.63 (*p* = 0.03) ↔GCF in the fully adjusted model (<70 years) β (*p* value) −0.03 (*p* = 0.79)			8
Nicoli et al., 2021, Italy [54]	Monzino 80-plus study	MDS	512	Non-cases: 62.8 Cases: 75.8	Non-cases: 91.9 ± 5.2 Cases: 92.1 ± 5.5	3.6			↔Dementia incidence in the fully adjusted model HR (95% CI) T1 (Ref) T2 1.17 (0.82, 1.66) T3 1.20 (0.82, 1.76)	7
Nishi et al., 2021, Spain [36]	NA (23 Spanish health centers)	MDS: 0–14	Baseline: 6647 Analysis: 5714	Baseline: 48%	Baseline: 65.0 ± 4.10	2	↑Cognitive function in the fully adjusted model β (95% CI; *p*-trend) ▪ ↑MMSE 0.070 (0.014, 0.175; *p*-trend = 0.011) ▪ ↓TMT-a −0.054 (−0.11, −0.002; *p*-trend = 0.047) ▪ ↓TMT-b −0.079 (−0.134, −0.024; *p*-trend = 0.004) ▪ ↔GCF, CDT, VFT-a, VFT-p, DST-f, DST-B			7
Corley et al., 2021, Scotland [55]	Lothian Birth Cohort 1936 study	MDP	863	49.7	69.5 ± 0.8	12.5 ± 0.5	↑Verbal ability in the fully adjusted model β (SE, *p* value) −0.003(0.001, *p* = 0.008)			8
Agarwal et al., 2021, USA [56]	CHAP	MDP	5001	63	74 ± 6.0	6.3 ± 2.8	↑Cognitive function in the fully adjusted model comparing highest vs. lowest intake β (95%CI) T1 (Ref) T2 0.014 (0.003, 0.025) T3 0.022 (0.010, 0.033)			7
Nooyens et al., 2021, The Netherlands [57]	Doetinchem Cohort Study	mMDS	3644	51	56 ± 7	15	↑GCF in the fully adjusted model comparing highest vs. lowest intake Mean (95% CI) 7.4% (1.0, 14.9%)			8
Charisis et al., 2021, Greece [58]	HELIAD study	MDP	1046	60	73.1 ± 5	3.1 ± 0.9			↓Dementia incidence in the fully adjusted model comparing highest vs. lowest intake HR (95% CI) T1 1 (Ref) T2 0.71 (0.36, 1.40) T3 0.75 (0.39, 1.43) T4 0.28 (0.10, 0.76)	8
Andreu-Reinón et al., 2021, Spain [59]	EPIC-Spain Dementia Cohort study	rMDS	16,160	59	30–70	21.6 ± 3.4			↓Dementia incidence for per 2-point increase in rMDS in the fully adjusted model comparing highest vs. lowest intake HR (95% CI; *p*-trend) 0.92 (0.85, 0.99; *p*-trend = 0.021)	9
Munoz-Garcia et al., 2020, Spain [37]	SUN cohort study	MDS: 0–14	806	30.3	61 ± 6	6 ± 3	↔Cognitive function for STICS-m score change in the fully adjusted model comparing highest vs. lowest intake OR (95% CI) T1 0 (Ref) T2 0.28 (−0.25, 0.80) T3 0.43 (−0.40, 1.26)			7
Hu et al., 2020, USA [60]	ARIC study	aMDS	13,630	56	54 ± 6	27			↔Dementia incidence in the fully adjusted model comparing highest vs. lowest intake HR (95% CI) Q1 1 (Ref) Q2 1.04 (0.90, 1.20) Q3 1.02 (0.88, 1.17) Q4 0.99 (0.86, 1.15) Q5 1.01 (0.88, 1.16)	9
Keenan et al., 2020, USA [61]	AREDS, AREDS2	aMDS	7756	AREDS: 68.7 AREDS2: 57.8	AREDS: 68.7 ± 4.9 AREDS2: 72.9 ± 7.7	5–10	↔Cognitive function in the fully adjusted model comparing highest vs. lowest intake			7
Hosking et al., 2019, Australia [38]	PATH study	9-point MDS: 0–9 Greek MDS: 0–50	1220	9-point MDS T1: 53 T2: 51 T3: 47 Greek MDS T1: 45 T2: 53 T3: 53	9-point MDS T1: 62.5 ± 1.5 T2: 62.4 ± 1.4 T3: 62.5 ± 1.5 Greek MDS T1: 62.3 ± 1.4 T2: 62.5 ± 1.5 T3: 62.5 ± 1.5	12		↔Cognitive impairment in the fully adjusted model comparing highest vs. lowest intake ▪ 9-point MDS T1 1 (Ref) T2 0.87 (0.47, 1.62) T31.30 (0.79, 2.15) ▪ Greek MDS T1 1 (Ref) T2 0.77 (0.45, 1.30) T3 0.77 (0.43, 1.39)		7
Shannon et al., 2019, UK [62]	EPIC-Norfolk Study	Pyramid MDS	8009	56	40–92	13–18	↑Cognitive function in the fully adjusted model comparing highest vs. lowest intake ▪↑Global cognition β (SE; *p* value) −0.012 (0.002; *p* < 0.001) ▪↑Verbal episodic memory −0.009 (0.002; *p* < 0.001) ↑Simple processing speed −0.002 (0.001; *p* = 0.013) ▪↑Verbal episodic memory OR (95% CI; *p* value) 0.784 (0.641, 0.959; *p* = 0.018) ▪↑Complex processing speed 0.739 (0.601, 0.907; *p* = 0.004) ▪↑Prospective memory 0.841 (0.724, 0.977; *p* = 0.023)			8
Mattei et al., 2019, USA [63]	BPRHS	MDS	557	73.6	56.0 ± 7.7	2	↑Memory function in the fully adjusted model β (SE; *p* value) 0.047 (0.02; *p* = 0.016)			7
Wu et al., 2019, Singapore [64]	SCHS	aMDS	16,948	59.2	53.5 ± 6.2	19.7		↓Cognitive impairment in the fully adjusted model comparing highest vs. lowest intake OR (95% CI) Q1 1 (Ref) Q2 0.85 (0.75, 0.96) Q3 0.75 (0.66, 0.86) Q4 0.67 (0.59, 0.77)		7
Shakersain et al., 2018, Sweden [41]	SNAC-K	MDS	2223	60.8	Men: 69.5 ± 8.6 Women: 71.3 ± 9.1		↑Cognitive function in the fully adjusted model β (95% CI; *p* value) ▪ MMSE continuous score 0.006 (0.002, 0.009; *p* = 0.002) Moderate intake 0.063 (−0.002, 0.129; *p* = 0.057) High intake 0.099 (0.036, 0.163; *p* = 0.002)			8
Tanaka et al., 2018, Italy [65]	InCHIANTI study	MDS	1139	56.5	75.4 ± 7.6	10.1	↑Cognitive function in the fully adjusted model comparing highest vs. lowest intake HR (95% CI; *p* value) 0.59 (0.39, 0.88; *p* = 0.011)			8
Bhushan et al., 2018, USA [66]	HPFS	aMDS	51,529	0	40–75	26	↑Cognitive function in the fully adjusted model comparing highest vs. lowest intake OR (95% CI) Q1 1 (Ref) Q2 0.95 (0.81, 1.10) Q3 0.74 (0.64, 0.86) Q4 0.67 (0.57, 0.78) Q5 0.64 (0.55, 0.75)			7
Richard et al., 2018, USA [67]	RBS of Healthy Aging study	aMDS	1499	58	73.2 ± 9.2	9 ± 7.7	↑Cognitive function in the fully adjusted model comparing highest vs. lowest intake ▪MMSE β (95% CI) T1 (Ref) T2 0.19 (−0.006, 0.38) T3 0.33 (0.11, 0.55)			8
Larsson et al., 2018, Sweden [68]	SIMPLER study	aMDS	28,775	47	71.6 ± 4.5	12.6			↔Dementia incidence in the fully adjusted model comparing highest vs. lowest intake HR (95% CI) Q1 1 (Ref) Q2 1.03 (0.88, 1.21) Q3 1.11 (0.95, 1.31) Q4 1.12 (0.96, 1.31)	8
Haring et al., 2016, USA [69]	WHIMS	aMDS	6425	100	65–79	9.11		↔MCI in the fully adjusted model comparing highest vs. lowest intake HR (95% CI) Q1 1 (Ref) Q2 1.26 (0.94, 1.68) Q3 1.08 (0.80, 1.46) Q4 0.98 (0.70, 1.35) Q5 0.82 (0.59, 1.14)	↔Dementia incidence in the fully adjusted model comparing highest vs. lowest intake HR (95% CI) Q1 1 (Ref) Q2 0.97 (0.67, 1.40) Q3 1.47 (1.05, 2.06) Q4 1.07 (0.73, 1.56) Q5 1.13 (0.79, 1.63)	7
Olsson et al., 2015, Sweden [70]	Uppsala longitudinal study	mMDS 0–8	1038	0	71 ± 0.6	Median: 11.6		↔Cognitive impairment in the fully adjusted model comparing highest vs. lowest intake OR (95% CI) T1 (Ref) T2 1.32 (0.82, 2.15) T3 0.64 (0.31, 1.30)	↔Dementia incidence in the fully adjusted model comparing highest vs. lowest intake HR (95% CI) T1 (Ref) T2 1.05 (0.67, 1.66) T3 0.85 (0.44, 1.62)	8
Galbete et al., 2015, Spain [71]	SUN cohort study	aMDS	823	27	61.9 ± 6.0	4	↑Cognitive function in the fully adjusted model comparing highest vs. lowest intake β (95% CI; *p* value) −0.56 (−0.99, −0.13; *p* = 0.01)			5
Gardener et al., 2015, Australia [72]	AIBL study	AusMDS	527	60.2	69.3 ± 6.4	3	↓Executive function cognitive domain APOE in ε4 allele carriers			6
Trichopoulou et al., 2015, Greece [73]	EPIC- Greece study	MDS	401	64	≥65	6.6	↑Cognitive function in the fully adjusted model comparing highest vs. lowest intake ▪Mildly lower MMSE score OR (95% CI; *p* value) T1 (Ref) T2 0.75 (0.41, 1.37; *p* = 0.348) T3 0.46 (0.25, 0.87; *p* = 0.017) ▪Substantially lower MMSE score T1 (Ref) T2 0.72 (0.31, 1.65; *p* = 0.441) T3 0.34 (0.13, 0.89; *p* = 0.029)			9
Koyama et al., 2015, USA [74]	Health ABC study	MDS	2326	51.3	74.6 ± 2.9	8	↑Cognitive function in MMSE points per year in the fully adjusted model MD (95% CI; *p* value) 0.22 (0.05, 0.39; *p* = 0.01)			9
Qin et al., 2015, China [75]	CHNS	aMDS	1650	Low intake: 52 Medium intake: 52 High intake: 47	Low intake: 64.0 Medium intake: 63.6 High intake: 62.9	5	↑Cognitive function per 1-point increase in aMDS in the fully adjusted model comparing highest vs. lowest intake β (95% CI) Low intake: 0 (Ref) Medium intake: 0.13 (−0.11, 0.38) High intake: 0.28 (0.02, 0.54) ↑Cognitive function in campsite scores per 1-point increase in aMDS in the fully adjusted model comparing highest vs. lowest intake β (95% CI) Low intake: 0 (Ref) Medium intake: 0.018 (−0.019, 0.056) High intake: 0.042 (0.002, 0.081)			9
Tangney et al., 2014, USA [76]	MAP	MDS	826	74	81.5 ± 7.1	4.1	↑Cognitive function β = 0.002, SEE = 0.001, *p* = 0.01)			5
Samieri et al., 2013, USA [77]	NHS	aMDS	16,058	100	74.3 ± 2.3	13	↑Cognitive function MDs (95% CI) ▪TICS Q1 (Ref) Q2 0.02 (20.02, 0.07) Q3 0.03 (20.01, 0.08) Q4 0.06 (0.02, 0.11) Q5 0.06 (0.01, 0.11) ▪Global score Q1 (Ref) Q2 0.02 (20.01, 0.05) Q3 0.03 (20.00, 0.06) Q4 0.04 (0.01, 0.07) Q5 0.05 (0.01, 0.08) ▪Verbal memory score Q1 (Ref) Q2 0.01 (20.03, 0.04) Q3 0.03 (−0.01, 0.06) Q4 0.04 (0.01, 0.08) Q5 0.06 (0.03, 0.10)			7
Samieri et al., 2013, USA [78]	Women’s Health Study	aMDS	6174	100	71.9 ± 4.1	4	↔Cognitive function			6
Tsivgoulis et al., 2013, USA [79]	REGARDS	MDS	17,478	57	64.4 ± 9.1	4.0 ± 1.5		↓Cognitive impairment OR (95% CI; *p* value) 0.87 (0.76, 1.00; *p* = 0.046)		7
Wengreen et al., 2013, USA [80]	CCMS	MDS	716	57	74 ± 9.7	11	↑Cognitive function 3MS Means ± SEs Q2 0.68 ± 0.29 Q3 0.62 ± 0.29 Q4 0.83 ± 0.29 Q5 0.94 ± 0.29 (P-quintile 5 compared with 1 = 0.0014)			8
Kesse-Guyot et al., 2013, France [81]	SU.VI.MAX study	MDS, MSDPS	3083	46	65.4 ± 4.6	13	↑Cognitive performance ▪Backward digit span ·MDS MD (95% CI) Q1 (Ref) Q2 0.03 (−0.81, 0.86) Q3 −0.64 (−1.60, 0.32) ▪Phonemic fluency task ·MSDPS Q1 (Ref) Q2 −0.61 (−1.45, 0.22) Q3 −1.00 (−1.85, −0.15)			8
Titova et al., 2013, Sweden [82]	PIVUS	MDS	194	50	70.1 ± 0.01	5	↔Cognitive function for 7MS in the fully adjusted model comparing highest vs. lowest intake β (*p* value) 0.11 (*p* = 0.13)			8
Vercambre et al., 2012, USA [83]	WACS	MDS	1557	100	66.1–91.2	5.4	↔Cognitive function ▪Global composite MD (95% CI) 0.01 (−0.01, 0.02)			6
Cherbuin et al., 2012, Australia [84]	PATH study	MDS	1528	51	60–69	4		↔MCI OR (95%CI) 1.41 (0.95, 2.10)	↔CDR OR (95% CI) 1.18 (0.88, 1.57)	6
Tangney et al., 2011, USA [85]	CHAP	MDS	3790	61.7	75.4 ± 6.2	7.6	↑Cognitive function β (SEE; *p* value) 0.0014 (0.0004, *p* = 0.0004)			7
Féart et al., 2009, France [86]	3C study	MDS	1410	60	75.9	4.1	↑Cognitive function in the fully adjusted model comparing highest vs. lowest intake β (95% CI; *p* value) ▪MMSE errors −0.03 (−0.05, −0.001; *p* = 0.04) ▪FCSRT 0.21 (0.008, 0.41; *p* = 0.04)		↔Dementia incidence in the fully adjusted model comparing highest vs. lowest intake HR (95% CI; *p* value) 1.12 (0.60, 2.10; *p* = 0.72)	8
Scarmeas et al., 2009, USA [87]	WHICAPstudy	MDP	1393	69	76.7 ± 6.58	4.5 ± 2.7		↔MCI in the fully adjusted model comparing highest vs. lowest intake HR (95% CI; *p* value) 0.72 (0.52 1.00; *p* = 0.05)		8
Psaltopoulou et al., 2008, Greece [88]	EPIC-Greece study	MDS	732	62	20–86	6–13	↔Cognitive function MMSE score β (95%CI) 0.05 (−0.09, 0.19)			7

3C study, Three-City study; 3MS, Modified Mini-Mental State Examination; 7MS, seven-minute screening; AIBL, Australian Imaging, Biomarkers and Lifestyle study of aging; aMDS, alternate Mediterranean diet score; APOE, apolipoprotein E; AREDS, Age-Related Eye Disease Study; ARIC, Atherosclerosis Risk in Communities; AusMDS, Australian-style Mediterranean diet score; BPRHS, Boston Puerto Rican Health Study; B-SEVLT, Brief Spanish-English Verbal Learning Test; CCMS, Cache County Memory Study; CDR, Clinical Dementia Rating; CDT, clock-drawing test; CHAP, Chicago Health and Aging Project; CHNS, China Health and Nutrition Survey; CI, confidence interval; DSST, Digit Symbol Substitution Test; DST-B, Digit Span Test—backward; DST-f, Digit Span Test—forward; EPAD LCS, European Prevention of Alzheimer’s Dementia Longitudinal Cohort Study; EPIC, European Prospective Investigation into Cancer and Nutrition; FCSRT, Free and Cued Selective Reminding Test; FMT, Four Mountains Test; GCF, global cognitive function; HCHS/SOL, Hispanic Community Health Study/Study of Latinos; Health ABC, Health, Aging, and Body Composition; HELIAD, Hellenic Epidemiological Longitudinal Investigation of Aging and Diet; HPFS, Health Professionals’ Follow-up Study; HR, hazard ratio; HRS, Health and Retirement Study; InCHIANTI, Invecchiare in Chianti, aging in the Chianti area; MAP, Memory and Aging Project; MAS, Sydney Memory and Ageing Study; MCI, mild cognitive impairment; MD, mean difference; MDCS, Malmö Diet and Cancer study; MDP, Mediterranean-dietary pattern; MDS, Mediterranean diet score; MED, Mediterranean; MEDAS, Mediterranean Diet Adherence Screener; mMDS, modified Mediterranean diet score; MMSE, Mini-Mental State Examination; MSDPS, Mediterranean-Style Dietary Pattern Score; MSLS, Maine–Syracuse Longitudinal Study; n, number; NA, not available; NHS, Nurses’ Health Study; OR, odds ratio; PAL, paired-associates learning; PATH, Personality and Total Health Through Life Cohort; PIVUS, Prospective Investigation of the Vasculature in Uppsala Seniors; PYRAMID, Mediterranean diet pyramid; Q, quintile; RBS, Rancho Bernardo Study; Ref, Reference; REGARDS, REasons for Geographic and Racial Differences in Stroke; rMDS, relative Mediterranean diet score; RR, relative risk; SCD, subjective cognitive decline; SCHS, Singapore Chinese Health Study; SD, standard deviation; SE, standard error; SEE, standard error of estimate; SIMPLER, Swedish Infrastructure for Medical Population-based Life-course Environmental Research, previously the Swedish Mammography Cohort and the Cohort of Swedish Men; SNAC-K, Swedish National study on Aging and Care in Kungsholmen; SOL–INCA, Latinos–Investigation of Neurocognitive Aging; STICS-m, Spanish Telephone Interview for Cognitive Status; SU.VI.MAX, Supplementation with Vitamins and Mineral Antioxidants; SUN, Seguimiento Universidad de Navarra; T, tertile; TICS, Telephone Interview for Cognitive Status; TMT-a, Trail Making Test, part a; TMT-b, Trail Making Test, part b; TwinsUK, United Kingdom Adult Twin Registry; UK, United Kingdom; USA, United States of America; VFT-a, verbal fluency tasks—semantical; VFT-p, verbal fluency tasks—phonological; WACS, Women’s Antioxidant Cardiovascular Study; WHICAP, Washington Heights–Inwood Columbia Aging Project; WHIMS, Women’s Health Initiative Memory Study; β, beta coefficient; ↑, significant increase in outcome; ↓, significant decrease in outcome; ↔, no significant effect.

**Table 5 nutrients-17-03469-t005:** Summary of the 17 publications (prospective studies) that investigated associations between DASH diet and cognitive outcomes.

Author, Year, Region	Study Name	Adherence	Subjects			Study Period (Follow-Up Years)	Outcomes			Study Quality
			Total (*n*)	Female (%)	Age (Range or Mean/SD or Median) (Years)	Average Follow-Up (Year)	Cognitive Function	Cognitive Impairment or MCI	Dementia	
Seago et al., 2024, USA [19]	HRS	DASH diet score	6154	60	69 ± 10	8	↑Cognitive function in the fully adjusted model β (95% CI; *p* value) 0.04 (0.01, 0.07; *p* = 0.004)			6
Bhave et al., 2024, USA [20]	REGARDS	DASH diet score	14,175	Non-cases: 57.9 Cases: 59.6	Non-cases: 63.4 ± 8.6 Cases: 65.8 ± 8.8	Non-cases: 10.9 Cases: 7.5		↓Cognitive impairment in the fully adjusted model HR (95% CI; *p* value) 0.96 (0.95, 0.98; *p* < 0.00005)		8
Chen et al., 2022, Australia [49]	MAS	DASH diet score	1037	55.2	78.8 ± 4.8	6	↔Global cognition in the fully adjusted model β (95% CI; *p* value) −0.001 (−0.010, 0.008; *p* = 0.781)			8
Yuan et al., 2022, USA [50]	NHS	DASH diet score	49,493	100	48 ± 7	31	↑Cognitive function in the fully adjusted model comparing highest vs. lowest intake ▪Moderate SCD OR (95% CI) Q1 1.00 (Ref) Q2 1.00 (0.94, 1.06) Q3 0.91 (0.86, 0.97) Q4 0.92 (0.86, 0.98) Q5 0.76 (0.71, 0.82) ▪Severe SCD Q1 1.00 (Ref) Q2 0.93 (0.84, 1.02) Q3 0.76 (0.68, 0.84) Q4 0.77 (0.69, 0.85) Q5 0.61 (0.55, 0.68)			7
Nishi et al., 2021, Spain [36]	NA (23 Spanish health centers)	DASH diet score: 8–40	baseline: 6647 analysis: 5714	48	65.0 ± 4.11	2	↔Cognitive function for MMSE, GCF, CDT, VFT-a, VFT-p, TMT-a, TMT-b, DST-B, DST-f in the fully adjusted model			7
Daniel et al., 2021, USA [89]	MESA cohort study	DASH diet score	4169	52.9	60.4 ± 9.5	2	↔Cognitive function in the fully adjusted model			6
Tong et al., 2021, Singapore [90]	SCHS	DASH diet score	14,683	59.1	72.9 ± 6.3	3		↓Cognitive impairment in the fully adjusted model comparing highest vs. lowest intake OR (95% CI) Q1 1.00 (Ref) Q2 0.82 (0.70, 0.96) Q3 0.65 (0.55, 0.76) Q4 0.67 (0.56, 0.80) Q5 0.50 (0.42, 0.59)		9
Munoz-Garcia et al., 2020, Spain [37]	SUN cohort study	DASH diet score: 8–40	806	30.3	61 ± 6	6 ± 3	↔Cognitive function for STICS-m score change in the fully adjusted model comparing highest vs. lowest intake β (95% CI) Q1 0 (Ref) Q2 −0.01 (−0.63, 0.60) Q3 −0.23 (−0.84, 0.38) Q4 −0.07 (−0.72, 0.58) Q5 0.30 (−0.35, 0.96)			7
Hu et al., 2020, USA [60]	ARIC study	DASH diet score	13,630	56	54 ± 6	27			↔Dementia incidence in the fully adjusted model comparing highest vs. lowest intake HR (95% CI) 1.10 (0.96, 1.26)	9
Mattei et al., 2019, USA [63]	BPRHS	DASH diet score	557	73.6	56.0 ± 7.7	2	↑Memory function in the fully adjusted model β (SE; *p* value) 0.24 (0.008; *p* = 0.003) ↑Word list learning 0.224 (0.097; *p* = 0.021) ↑Stroop 0.271 (0.091; *p* = 0.003)			7
Wu et al., 2019, Singapore [64]	SCHS	DASH diet score	16,948	59.2	53.5 ± 6.2	19.7		↓Cognitive impairment in the fully adjusted model comparing highest vs. lowest intake OR (95% CI) Q1 1.00 (Ref) Q2 0.84 (0.74, 0.95) Q3 0.73 (0.64, 0.83) Q4 0.71 (0.62, 0.81)		7
Shakersain et al., 2018, Sweden [41]	SNAC-K	DASH diet score	2223	60.8	Men: 69.5 ± 8.6 Women: 71.3 ± 9.1	6	↔MMSE in the fully adjusted model			8
Larsson et al., 2018, Sweden [68]	SIMPLER study	DASH diet score	28,775	47	71.6 ± 4.5	12.6			↔Dementia incidence in the fully adjusted model comparing highest vs. lowest intake HR (95% CI) Q1 1.00 (Ref) Q2 0.96 (0.88, 1.06) Q3 0.94 (0.85, 1.03) Q4 0.96 (0.87, 1.05)	8
Berendsen et al., 2017, USA [91]	NHS	DASH diet score	16,144	100	74.3± 2.3	4.1	↑Global cognitive score in the fully adjusted model comparing highest vs. lowest intake Mean (95% CI) Q1 (Ref) Q2 0.02 (−0.01, 0.05) Q3 0.01 (−0.02, 0.04) Q4 0.03 (0.00, 0.06) Q5 0.04 (0.01, 0.07) ↑Verbal memory score in the fully adjusted model comparing highest vs. lowest intake Q1 (Ref) Q2 0.02 (−0.01, 0.05) Q3 0.00 (−0.03, 0.04) Q4 0.03 (0.00, 0.07) Q5 0.04 (0.01, 0.07) ↑TICS score in the fully adjusted model comparing highest vs. lowest intake Q1 (Ref) Q2 0.10 (−0.03, 0.22) Q3 0.08 (−0.05, 0.20) Q4 0.09 (−0.04, 0.22) Q5 0.16 (0.03, 0.29)			6
Haring et al., 2016, USA [69]	WHIMS	DASH diet score	6425	100	65–79	9.11		↔MCI in the fully adjusted model comparing highest vs. lowest intake HR (95% CI) Q1 1 (Ref) Q2 0.94 (0.69, 1.28) Q3 0.98 (0.81, 1.36) Q4 0.82 (0.60, 1.12) Q5 0.72 (0.52, 1.02)	↔Dementia incidence in the fully adjusted model comparing highest vs. lowest intake HR (95% CI) Q1 1 (Ref) Q2 1.12 (0.75, 1.66) Q3 1.17 (0.77, 1.76) Q4 1.40 (0.96, 2.05) Q5 1.28 (0.86, 1.91)	7
Tangney et al., 2014, USA [76]	MAP	DASH diet score	826	74	81.5 ± 7.1	4.1	↑Cognitive function in the fully adjusted model β (SEE; *p* value) 0.007 (0.003; *p* = 0.03)			5
Wengreen et al., 2013, USA [80]	CCMS	DASH diet score	716	57	74 ± 9.7	11	↑Cognitive function in the fully adjusted model comparing highest vs. lowest intake ▪3MS Means ± SEs Q2 0.35 ± 0.29 Q3 0.68 ± 0.29 Q4 0.96 ± 0.29 Q5 0.97 ± 0.29 (P-quintile 5 compared with 1)			8

3MS, Modified Mini-Mental State Examination; ARIC, Atherosclerosis Risk in Communities; BPRHS, Boston Puerto Rican Health Study; CCMS, Cache County Memory Study; CDT, clock-drawing test; CI, confidence interval; DASH, Dietary Approaches to Stop Hypertension; DST-B, Digit Span Test—backward; DST-f, Digit Span Test—forward; GCF, global cognitive function; HR, hazard ratio; HRS, Health and Retirement Study; MAP, Memory and Aging Project; MAS, Sydney Memory and Ageing Study; MCI, mild cognitive impairment; MESA, Multi-Ethnic Study of Atherosclerosis; MMSE, Mini-Mental State Examination; n, number; NA, not available; NHS, Nurses’ Health Study; OR, odds ratio; Q, quintile; Ref, reference; REGARDS, REasons for Geographic and Racial Differences in Stroke; SCD, subjective cognitive decline; SCHS, Singapore Chinese Health Study; SD, standard deviation; SE, standard error; SEE, standard error of estimate; SIMPLER, Swedish Infrastructure for Medical Population-based Life-course Environmental Research, previously the Swedish Mammography Cohort and the Cohort of Swedish Men; SNAC-K, Swedish National study on Aging and Care in Kungsholmen; STICS-m, Spanish Telephone Interview for Cognitive Status; SUN, Seguimiento Universidad de Navarra; TICS, Telephone Interview for Cognitive Status; TMT-a, Trail Making Test, part a; TMT-b, Trail Making Test, part b; USA, United States of America; VFT-a, verbal fluency tasks—semantical; VFT-p, verbal fluency tasks—phonological; WHIMS, Women’s Health Initiative Memory Study; β, beta coefficient; ↑, significant increase in outcome; ↓, significant decrease in outcome; ↔, no significant effect.

**Table 6 nutrients-17-03469-t006:** Summary of the 11 publications (prospective studies) that investigated associations between HEI and cognitive outcomes.

Author, Year, Region	Study Name	Adherence	Subjects			Study Period (Follow-Up Years)	Outcomes			Study Quality
			Total (*n*)	Female (%)	Age (Range or Mean/SD or Median) (Years)	Average Follow-Up (Year)	Cognitive Function	Cognitive Impairment or MCI	Dementia	
Cornelis et al., 2022, UK [30]	UK Biobank study	AHEI-2010 score	120,661	56.5	T1: 56.9 ± 8.1 T2: 58.1 ± 7.8 T3: 58.6 ± 7.6	10.5	↑Cognitive function in the fully adjusted model comparing highest vs. lowest intake ▪FI test β (95% CI; *p* value) T1 (Ref) T2 −0.05 (−0.09, −0.008; *p* = 0.02) T3 −0.17 (−0.21, −0.13; *p* < 0.0001) ▪Reaction Time T1 (Ref) T2 1.23 (−0.12, 2.57; *p* = 0.07) T3 2.77 (1.37, 4.16; *p* < 0.0001) ▪Pairs matching test T1 (Ref) T2 0.03 (0.02, 0.04; *p* < 0.0001) T3 0.04 (0.03, 0.05; *p* < 0.0001) ▪SDS test T1 (Ref) T2 −0.19 (−0.27, −0.11; *p* < 0.0001) T3−0.40 (−0.49, −0.32; *p* < 0.0001) ▪Trail A test T1 (Ref) T2 0.009 (0.003, 0.01; *p* = 0.002) T3 0.02 (0.01, 0.03; *p* < 0.0001) ▪Trail B test T1 (Ref) T2 0.015 (0.009, 0.021; *p* < 0.0001) T3 0.034 (0.028, 0.039; *p* < 0.0001) ▪Prospective Memory Test T1 (Ref) T2 0.89 (0.84, 0.95; *p* = 0.0003) T3 0.90 (0.85, 0.96; *p* = 0.002)		↔Dementia incidence in the fully adjusted model comparing highest vs. lowest intake HR (95% CI; *p* value) T1 (Ref) T2 0.93 (0.78, 1.10; *p* = 0.38) T3 0.89 (0.75, 1.06; *p* = 0.20)	9
Yuan et al., 2022, USA [50]	NHS	AHEI-2010 score: 0–110	49,493	100	48 ± 7	31	↑Cognitive function in the fully adjusted model comparing highest vs. lowest intake ▪Moderate SCD OR (95% CI) Q1 1.00 (Ref) Q2 0.97 (0.92, 1.04) Q3 0.99 (0.93, 1.06) Q4 0.93 (0.87, 0.99) Q5 0.93 (0.87, 0.99) ▪Severe SCD Q1 1.00 (Ref) Q2 0.88 (0.80, 0.96) Q3 0.90 (0.82, 0.99) Q4 0.84 (0.76, 0.93) Q5 0.81 (0.73, 0.90)			7
Munoz-Garcia et al., 2020, Spain [37]	SUN cohort study	AHEI-2010 score: 0–110	806	30.3	61 ± 6	6 ± 3	↑Cognitive function for STICS-m score change in the fully adjusted model comparing highest vs. lowest intake β (95% CI) Q1 0 (Ref) Q2 0.43 (−0.18, 1.04) Q3 0.42 (−0.23, 1.07) Q4 0.30 (−0.33, 0.93) Q5 0.81 (0.17, 1.45; *p* < 0.05) ↑Cognitive function for each 9 points (0–110) in the fully adjusted model β (95% CI; *p* value) 0.25 (0.04, 0.45; *p* < 0.05)			7
Hu et al., 2020, USA [60]	ARIC study	AHEI-2010 score	13,630	56	54 ± 6	27			↔Dementia incidence with AHEI-2010 in the fully adjusted model comparing highest vs. lowest intake HR (95% CI) 1.04 (0.91, 1.20)	9
		HEI-2015 score							↓Dementia incidence with HEI-2015 in the fully adjusted model comparing highest vs. lowest intake HR (95% CI) 0.86 (0.74, 0.99)	
Mattei et al., 2019, USA [63]	BPRHS	AHEI-2010 score	557	73.6	56.0 ± 7.7	2	↑Memory function in the fully adjusted model 0.012 (0.004; *p* = 0.001) ↑Word recognition in the fully adjusted model 0.062 (0.021; *p* = 0.004)			7
		HEI-2005 score					▪HEI-2005 ↑Memory function in the fully adjusted model β (SE; *p* value) 0.011 (0.003; *p* = 0.002) ↑Word recognition in the fully adjusted model 0.063 (0.02; *p* = 0.002)			
Wu et al., 2019, Singapore [64]	SCHS	AHEI-2010 score	16,948	59.2	53.5 ± 6.2	19.7		↓Cognitive impairment in the fully adjusted model comparing highest vs. lowest intake OR (95% CI) Q1 1 (Ref) Q2 0.87 (0.77, 0.99) Q3 0.80 (0.70, 0.90) Q4 0.75 (0.66, 0.85)		7
Akbaraly et al., 2019, UK [92]	WII	AHEI-2010 score: 0–110	8225	30.9	50.2	24.8	↔Cognitive function for 18 years between per 1-SD increase in HFDP in the fully adjusted model β (95% CI; *p* value) 0.01 (−0.01, 0.03; *p* = 0.23)		↔Dementia incidence in the fully adjusted model comparing highest vs. lowest intake HR (95% CI) ▪AHEI in 1991–1993 T1 1 (Ref) T2 0.95 (0.73, 1.23) T3 0.93 (0.71, 1.22) Per 1-SD (10-point) in increase: 0.97 (0.87, 1.08) ▪AHEI in 1997–1999 T1 1 (Ref) T2 0.98 (0.69, 1.38) T3 0.95 (0.67, 1.35) Per 1-SD (10-point) in increase: 0.97 (0.83, 1.12) ▪AHEI in 2002–2004 T1 1 (Ref) T2 0.81 (0.58, 1.15) T3 0.73 (0.51, 1.05) Per 1-SD (10-point) in increase: 0.87 (0.75, 1.00)	7
Richard et al., 2018, USA [67]	RBS of Healthy Aging study	AHEI-2010 score	1499	58	73.2 ± 9.2	9 ± 7.7	↔Cognitive function in the fully adjusted model comparing highest vs. lowest intake MMSE β (95% CI) T1 (Ref) T2 0.18 (−0.02, 0.37) T3 0.11 (−0.09, 0.31)			8
Haring et al., 2016, USA [69]	WHIMS	AHEI-2010 score: 0–110	6425	100	65–79	9.11		↔MCI in the fully adjusted model comparing highest vs. lowest intake HR (95% CI) Q1 1 (Ref) Q2 0.97 (0.73, 1.29) Q3 0.98 (0.72, 1.33) Q4 0.96 (0.71, 1.29) Q5 0.75 (0.54, 1.03)	↔Dementia incidence in the fully adjusted model comparing highest vs. lowest intake HR (95% CI) Q1 1 (Ref) Q2 1.05 (0.74, 1.48) Q3 1.22 (0.86, 1.75) Q4 1.28 (0.91, 1.81) Q5 1.01 (0.71, 1.46)	7
Shatenstein et al., 2012 Canada [93]	NuAge study	C-HEI	1488	52.6	men: 74.05 ± 4.09 women: 74.36 ± 4.21	3	↔Cognitive function in the fully adjusted model comparing highest vs. lowest intake			6
Tangney et al., 2011, USA [85]	CHAP	HEI-2005 score	3790	61.7	75.4 ± 6.2	7.6	↔Cognitive function in the fully adjusted model β (SEE; *p* value) 0.0002 (0.0002; *p* = 0.214)			7

AHEI, Alternative Healthy Eating Index; ARIC, Atherosclerosis Risk in Communities; BPRHS, Boston Puerto Rican Health Study; CHAP, Chicago Health and Aging Project; C-HEI, Canadian Healthy Eating Index; CI, confidence interval; FI, fluid intelligence; HEI, Healthy Eating Index; HFDP, Healthy Food Dietary Pattern; HR, hazard ratio; MCI, mild cognitive impairment; MMSE, Mini-Mental State Examination; n, number; NHS, Nurses’ Health Study; NuAge, Quebec Longitudinal Study on Nutrition and Successful Aging; OR, odds ratio; Q, quintile; RBS, Rancho Bernardo Study; Ref, reference; SCD, subjective cognitive decline; SCHS, Singapore Chinese Health Study; SD, standard deviation; SDS, symbol digit substitution; SE, standard error; SEE, standard error of estimate; STICS-m, Spanish Telephone Interview for Cognitive Status; SUN, Seguimiento Universidad de Navarra; T, tertile; UK, United Kingdom; USA, United States of America; WHIMS, Women’s Health Initiative Memory Study; WII, Whitehall II study; β, beta coefficient; ↑, significant increase in outcome; ↓, significant decrease in outcome; ↔, no significant effect.

**Table 7 nutrients-17-03469-t007:** Summary of the 8 publications (prospective studies) that investigated associations between plant-based patterns diet and cognitive outcomes.

Author, Year, Region	Study Name	Adherence	Subjects			Study Period (Follow-Up Years)	Outcomes			Study Quality
			Total (*n*)	Female (%)	Age (Range or Mean/SD or Median) (Years)	Average Follow-Up (Year)	Cognitive Function	Cognitive Impairment or MCI	Dementia	
Zhang et al., 2023, UK [26]	UK Biobank Study	hPDI	114,684	55.5	56.8 ± 7.77	9.4			↔Dementia incidence in the fully adjusted model comparing highest vs. lowest intake HR (95% CI; *p* value) T1 1 (Ref) T2 1.02 (0.81, 1.27; *p* = 0.88) T3 0.77 (0.77, 1.22; *p* = 0.78)	9
de Crom et al., 2023, The Netherlands [94]	Rotterdam Study	hPDI score	9543	58	64.1 ± 8.6	14.5			↓Dementia incidence with hPDI in men HR (95% CI) 0.86 (0.75, 0.99) ↓Dementia incidence with hPDI in APOE ε4 carriers 0.83 (0.73, 0.95)	9
van Soest et al., 2023, The Netherlands [95]	B-proof	hPDI	314	47	72.1 ± 5.4	2.0	↔GCF in the fully adjusted model comparing highest vs. lowest intake β (95% CI; *p* value) 0.05 (−0.03, 0.12; *p* = 0.21)			6
		uPDI					↔GCF in the fully adjusted model comparing highest vs. lowest intake β (95% CI; *p* value) −0.04 (−0.11, 0.04; *p* = 0.33)			
Wu et al., 2023, UK [96]	UK Biobank study	PDI	180,532	Q1: 52.3 Q3: 56.1 Q5: 56.2	Q1: 56.0 Q3: 57.0 Q5: 57.0	10			↔Dementia incidence in the fully adjusted model comparing highest vs. lowest intake HR (95% CI) 1.03 (0.87, 1.23)	9
		hPDI		Q1: 43.4 Q3: 56.4 Q5: 66.7	Q1: 54.0 Q3: 57.0 Q5: 58.0				↓Dementia incidence in the fully adjusted model comparing highest vs. lowest intake HR (95% CI) Q1 1 (Ref) Q2 0.98 (0.83, 1.17) Q3 0.88 (0.73, 1.05) Q4 0.80 (0.67, 0.96) Q5 0.82 (0.68, 0.98)	
		uPDI		Q1: 57.5 Q3: 55.6 Q5: 51.2	Q1: 58.0 Q3: 57.0 Q5: 53.0				↑Dementia incidence in the fully adjusted model comparing highest vs. lowest intake HR (95% CI) Q1 1 (Ref) Q2 0.96 (0.81, 1.16) Q3 1.05 (0.89, 1.23) Q4 1.21 (1.02, 1.45) Q5 1.29 (1.08, 1.53)	
Liu et al., 2022, USA [97]	CHAP	PDI	3337	64.0	73.7 ± 5.7	NA	↔Cognitive function in the fully adjusted model comparing highest vs. lowest intake			7
		hPDI					↑Cognitive function for African American subjects in the fully adjusted model comparing highest vs. lowest intake β (SE; *p* value) 0.0183 (0.0086; *p* = 0.032)			
		uPDI					↔Cognitive function in the fully adjusted model comparing highest vs. lowest intake			
Zhu et al., 2022, China [98]	CLHLS	PDI	6136	46.3	80 ± 9.83	10.0		↓Cognitive impairment in the fully adjusted model comparing highest vs. lowest intake OR (95% CI; *p* value) Q1 1 (Ref) Q2 0.90 (0.81, 1.01; *p* = 0.64) Q3 0.64 (0.57, 0.72; *p* < 0.001) Q4 0.45 (0.39, 0.52; *p* < 0.001)		8
		hPDI						↓Cognitive impairment in the fully adjusted model comparing highest vs. lowest intake OR (95% CI; *p* value) Q1 1 (Ref) Q2 0.90 (0.81, 1.00; *p* = 0.044) Q3 0.76 (0.67, 0.85; *p* < 0.001) Q4 0.61 (0.54, 0.70; *p* < 0.001)		
		uPDI						↑Cognitive impairment in the fully adjusted model comparing highest vs. lowest intake OR (95% CI; *p* value) Q1 1 (Ref) Q2 1.17 (1.03, 1.33; *p* = 0.014) Q3 1.47 (1.30, 1.66; *p* < 0.001) Q4 2.03 (1.79, 2.31; *p* < 0.001)		
Liang et al., 2022, China [99]	CLHLS	PDI	4792	49.4	80.70 ± 9.58	PY: 24,156		↓Cognitive impairment in the fully adjusted model comparing highest vs. lowest intake HR (95% CI; *p* value) 1.32 (1.16, 1.50; *p* < 0.001)		8
		hPDI						↓Cognitive impairment in the fully adjusted model comparing highest vs. lowest intake HR (95% CI; *p* value) 1.46 (1.29, 1.66; *p* < 0.001)		
		uPDI						↑Cognitive impairment in the fully adjusted model comparing highest vs. lowest intake HR (95% CI; *p* value) 1.21 (1.06, 1.38; *p* = 0.004)		
Wu et al., 2019, Singapore [64]	SCHS	PDI	16,948	59.2	53.5 ± 6.2	19.7		↓Cognitive impairment in the fully adjusted model comparing highest vs. lowest intake OR (95% Cl) Q1 1 (Ref) Q2 0.87 (0.77, 0.98) Q3 0.75 (0.66, 0.86) Q4 0.82 (0.71, 0.94)		7
		hPDI						↓Cognitive impairment in the fully adjusted model comparing highest vs. lowest intake OR (95% Cl) Q1 1 (Ref) Q2 0.88 (0.77, 1.00) Q3 0.85 (0.75, 0.97) Q4 0.78 (0.68, 0.90)		

APOE, apolipoprotein E; B-proof, the B-vitamins for the Prevention of Osteoporotic Fractures; CHAP, Chicago Health and Aging Project; CI, confidence interval; CLHLS, Chinese Longitudinal Healthy Longevity Survey; GCF, global cognitive function; hPDI, healthy plant-based dietary index; HR, hazard ratio; MCI, mild cognitive impairment; n, number; NA, not available; OR, odds ratio; PDI, overall plant-based dietary index; PY, person-year; Q, quintile; Ref, reference; SCHS, Singapore Chinese Health Study; SD, standard deviation; SE, standard error; T, tertile; UK, United Kingdom; uPDI, unhealthy plant-based dietary index; USA, United States of America; β, beta coefficient; ↑, significant increase in outcome; ↓, significant decrease in outcome; ↔, no significant effect.

**Table 8 nutrients-17-03469-t008:** Summary of the 33 publications (36 prospective studies) that investigated associations between healthy dietary patterns and cognitive outcomes.

Dietary Pattern	Author, Year, Region	Study Name	Adherence	Subjects			Study Period (Follow-Up Years)	Outcomes			Study Quality
				Total (*n*)	Female (%)	Age (Range or Mean/SD or Median) (Years)	Average Follow-Up (Year)	Cognitive Function	Cognitive Impairment or MCI	Dementia	
Dutch dietary guidelines	de Crom et al., 2022, The Netherlands [29]	Rotterdam Study	DDG score	Baseline I: 5375	Baseline I: 59.0	Baseline I: 67.7 ± 7.8	Baseline I: 15.6			↔Dementia incidence in the fully adjusted model comparing highest vs. lowest intake	9
				Baseline II: 2861	Baseline II: 57.4	Baseline II: 75.3 ± 5.9	Baseline II: 5.9			↔Dementia incidence in the fully adjusted model comparing highest vs. lowest intake	9
	Nooyens et al., 2021, The Netherlands [57]	Doetinchem Cohort Study	mDHD15-index	3644	51	56 ± 7	15	↑GCF in the fully adjusted model comparing highest vs. lowest intake Mean (95% CI) 6.5% (0.6, 13.6) ↑Cognitive flexibility in the fully adjusted model comparing highest vs. lowest intake Mean (95% CI) 10.3% (3.7, 18.3)			8
Australian Dietary Guidelines	Chen et al., 2021, Australia [110]	MAS	ADG	1037	55.2	78.8 ± 4.8	6	↔Global cognition in the fully adjusted model comparing highest vs. lowest intake β (95% CI) 0.000 (−0.007, 0.007)			7
	Milte et al., 2019, Australia [111]	WELL study	Diet quality (Australian DGI-2013)	617	51	60.2 ± 3.14	5	↑Cognitive function in the fully adjusted model in men β (95% CI) 0.03 (0.00, 0.07; *p* = 0.07)			5
Japanese diet pattern	Zhang et al., 2023, Japan [112]	NILS-LSA project	wJDI9 score: −1 to 12	1504	51	65–82	11.4			↓Dementia incidence in the fully adjusted model comparing highest vs. lowest intake HR (95% CI; *p* value) 0.56 (0.34, 0.93; *p* = 0.024)	8
	Lu et al., 2020, Japan [113]	Ohsaki Cohort Study and Ohsaki Cohort 2006 Study	JDI8 score	3146	54	≥65 years	5.0 ± 1.4			↓Dementia incidence in the fully adjusted model comparing highest vs. lowest intake HR (95% CI) ▪Great decreased JDI8 scores: 1.72 (1.13, 2.62) ▪Moderate decreased JDI8 scores: 1.10 (0.73, 1.66) ▪Great increased JDI8 0.62 (0.38, 1.02) ▪Moderate increased JDI8 0.82 (0.54, 1.25) (*p*-trend < 0.0001)	8
	Tomata et al., 2016, Japan [109]	Ohsaki Cohort 2006 Study	Japanese dietary pattern	14,402	56	73.8 ± 5.9	4.9 ± 1.5			↓Dementia incidence in the fully adjusted model comparing highest vs. lowest intake HR (95% CI) Q1 1 (Ref) Q2 0.95 (0.81, 1.11) Q3 0.85 (0.71, 1.01) Q4 0.80 (0.66, 0.97)	8
Nordic Prudent Dietary Pattern	Wu et al., 2021, Sweden [106]	SNAC-K	NPDP	2290	60.8	70.8 ± 9.1	10			↑Dementia incidence in the fully adjusted model comparing highest vs. lowest intake HR (95% CI) T1 1 (Ref) T2 1.02 (0.81, 1.12) T3 1.19 (1.04, 1.34)	8
	Shakersain et al., 2018, Sweden [41]	SNAC-K	NPDP	2223	60.8	men: 69.5 ± 8.6 women: 71.3 ± 9.1	6	↑Cognitive function for MMSE in the fully adjusted model comparing highest vs. lowest intake ▪Moderate intake β (95% CI; *p* value) 0.139 (0.077, 0.201; *p* < 0.001) ▪High intake 0.238 (0.175, 0.300; *p* < 0.001)			8
Baltic Sea Diet	Shakersain et al., 2018, Sweden [41]	SNAC-K	Baltic Sea Diet indices	2223	60.8	men: 69.5 ± 8.6 women: 71.3 ± 9.1	6	↔Cognitive function for MMSE in the fully adjusted model			8
Fruits and/or vegetables	Rivan et al., 2022, Malaysia [107]	LRGS-TUA	Tropical fruits-oats dietary pattern	280	48.6	67.3 ± 5	5		↔MCI in the fully adjusted model comparing highest vs. lowest intake OR (95% CI; *p* value) 1.728 (0.568, 5.258; *p* = 0.335)	↓Dementia incidence in the fully adjusted model comparing highest vs. lowest intake OR (95% CI; *p* value) 0.101 (0.011, 0.967; *p* = 0.047)	8
	Chen et al., 2017, Taiwan [114]	NA (elderly health checkup program at National Taiwan University Hospital, Taipei, Taiwan)	Vegetable dietary pattern	475	52	≥65	2	↑Cognitive function for Logical Memory-Recall I in the fully adjusted model comparing highest vs. lowest intake β (95% CI) T1 (Ref) T2 0.18 (0.03, 0.33) T3 0.16 (0.01, 0.32) OR (95% CI) T1 1.00 T2 0.48 (0.28, 0.83) T3 0.42 (0.24, 0.74) ↔Global cognition in the fully adjusted model comparing highest vs. lowest intake			5
	Ashby-Mitchell et al., 2015, Australia [115]	AusDiab study	Fruit and vegetable pattern	577	49.22	66.07 ± 4.85	3		↓Cognitive impairment in the fully adjusted model comparing highest vs. lowest intake OR (95% CI; *p* value) 1.061 (1.006, 1.118; *p* = 0.03)		6
	Titova et al., 2013, Sweden [82]	PIVUS	Vegetable & legumes	194	50	70.1 ± 0.01	5	↔Cognitive function for 7MS in the fully adjusted model comparing highest vs. lowest intake β (*p* value) 0.10 (*p* = 0.21)			8
Healthy dietary pattern	O’Reilly et al., 2024, Australia [16]	PATH study	DGI, IDQ	1753	Low: 45 Medium: 53 High: 57	60–64	12		↔Cognitive impairment in the fully adjusted model comparing highest vs. lowest intake ▪DGI OR (95% Cl) T1 (Ref) T2 0.69 (0.42, 1.11) T3 0.76 (0.48, 1.22) ▪IDQ OR (95% Cl) T1 (Ref) T2 0.99 (0.61, 1.62) T3 1.20 (0.73, 1.98)		7
	Rogers-Soeder et al., 2024, USA [100]	MrOS	PDP scores	4231	0	72 ± 5.5	4.6 ± 0.3	↔Cognitive function in the fully adjusted model comparing highest vs. lowest intake			6
	Flores et al., 2023, USA [116]	GRAS	Diet quality	2232	59	84 ± 3.7	6.9			↔Dementia incidence in the fully adjusted model comparing highest vs. lowest intake HR (95% CI) 1.01 (0.79, 1.29)	9
	Schulz et al., 2023, UK [117]	UK Biobank study	Diet score: A higher score means a healthier diet	104,895	54	57.1 ± 8.0	7.3	↔Cognitive function in the fully adjusted model comparing highest vs. lowest intake			7
	Zhang et al., 2023, China [118]	CHNS	“Vegetable-pork” dietary score	6308	52	≥55	22	↑Global cognitive score in the fully adjusted model comparing highest vs. lowest intake OR (95% CI) Q1 1 (Ref) Q2 0.82 (0.73, 0.93) Q3 0.79 (0.69, 0.91) Q4 0.74 (0.63, 0.86)			9
	Glans et al., 2023, Sweden [52]	MDCS	SDGS score: 0–5	28,025	60.7	58.1 ± 7.6	19.8			↔Dementia incidence in the fully adjusted model comparing highest vs. lowest intake	9
	Samuelsson et al., 2022, Sweden [101]	Gothenburg H70 birth cohort study	HDP	602	64	70.6 ± 0.3	12.8			↓Dementia incidence for APOE ε4 non-carriers in the fully adjusted model comparing highest vs. lowest intake HR (95% CI; *p* value) 0.77 (0.61, 0.98; *p* = 0.03)	9
	Nooyens et al., 2021, The Netherlands [57]	Doetinchem Cohort Study	HDI	3644	51	56 ± 7	15	↑GCF in the fully adjusted model comparing highest vs. lowest intake Mean (95% CI) 6.5% (0.3, 13.7)			8
	Parrott et al., 2021, Canada [102]	NuAge study	PDP	350	54	73.7 ± 3.8	4	↔GCF in the fully adjusted model comparing highest vs. lowest intake β (SE; *p* value) −0.06 (0.06; *p* = 0.339) ↔Executive function in the fully adjusted model comparing highest vs. lowest intake −0.01 (0.27; *p* = 0.984)			6
	Shang et al., 2021, China [119]	CHNS	HDP	2307	50.8	70.2 ± 6.9	7	↑Cognitive function in the fully adjusted model comparing highest vs. lowest intake OR (95% CI) 0.61 (0.42, 0.89)			9
	Akbaraly et al., 2019, UK [92]	WII	HFDP	8225	30.9	50.2	24.8	↑Cognitive function for 18 years between per 1-SD increase in HFDP in the fully adjusted model β (95% CI; *p* value) −0.03 (−0.05, −0.01; *p* = 0.007)		↔Dementia incidence in the fully adjusted model comparing highest vs. lowest intake ▪HFDP in 1991–1993 HR (95% CI) T1 1 (Ref) T2 1.01 (0.77, 1.34) T3 0.97 (0.73, 1.30) Per 1-SD (10-point) in increase: 0.93 (0.83, 1.05) ▪HFDP in 1997–1999 T1 1 (Ref) T2 0.95 (0.67, 1.35) T3 0.83 (0.56, 1.22) Per 1-SD (10-point) in increase: 0.86 (0.72, 1.02) ▪HFDP in 2002–2004 T1 1 (Ref) T2 0.88 (0.61, 1.25) T3 0.70 (0.47, 1.05) Per 1-SD (10-point) in increase: 0.90 (0.76, 1.07)	7
	Shakersain et al., 2016, Sweden [103]	SNAC-K	PDP	2223	60.8	70.6 ± 8.9	6	↑Cognitive function for MMSE in the fully adjusted model comparing highest vs. lowest intake β (95% CI; *p* value) 0.106 (0.024, 0.189; *p* = 0.011)			8
	Olsson et al., 2015, Sweden [70]	Uppsala longitudinal study	HDI	1038	0	71 ± 0.6	11.6		↔Cognitive impairment in the fully adjusted model comparing highest vs. lowest intake	↔Dementia incidence in the fully adjusted model comparing highest vs. lowest intake	8
			LCHP						↔Cognitive impairment in the fully adjusted model comparing highest vs. lowest intake	↔Dementia incidence in the fully adjusted model comparing highest vs. lowest intake	
	Gardener et al., 2015, Australia [72]	AIBL study	PD score	527	60.2	69.3 ± 6.4	3	↔Cognitive function for composite cognitive domain of APOE ε4 allele carriers in the fully adjusted model			6
	Tsai et al., 2015, Taiwan [104]	TLSA study	HDP	2988	45.7	73 ± 6	8	↔Cognitive function in the fully adjusted model comparing highest vs. lowest intake OR (95% CI) 1.13 (0.53, 2.41)			7
	Parrott et al., 2013, Canada [105]	NuAge study	PDP	1099	49.4	74.1 ± 4.1	3	↑Cognitive function comparing highest vs. lowest intake β (*p* value) β (95% CI; *p* value) PDP with high education 0.44 (0.080, 0.80; *p* = 0.017) PDP with high income 0.56 (0.11, 1.01; *p* = 0.015) PDP with high composite SEP 0.37 (0.045, 0.70; *p* = 0.026)			6
	Ozawa et al., 2013, Japan [121]	Hisayama study	DP high in soybeans, vegetables, algae, and milk and dairy and low in rice	1006	57	68	15			↓Dementia incidence in the fully adjusted model comparing highest vs. lowest intake HR (95% CI) Q1 1 (Ref) Q2 0.85 (0.61, 1.19) Q3 0.72 (0.50, 1.02) Q4 0.66 (0.46, 0.95)	8
Vegetarian diet	Fan et al., 2023, Taiwan [108]	HAICDDS Project	Vegetarian diet	1285	53	mean = 72.36	Mean follow-up duration = 2.33 years (days) Incident dementia = 428.07 ± 234.94 Without incident dementia = 1264.80 ± 437.34 Incident Alzheimer’s dementia = 425 ± 209.31 Incident vascular dementia = 425 ± 259.19			↑Dementia incidence in the fully adjusted model comparing highest vs. lowest intake HR (95% CI; *p* value) 1.95 (1.12, 4.30; *p* < 0.0001)	9
	Tsai et al., 2022, Taiwan [122]	TCVS	Taiwanese vegetarian diet	5710	63.1	57.8 ± 6.5	9.2			↓Dementia incidence in the fully adjusted model comparing highest vs. lowest intake HR (95% CI; *p* value) 0.671 (0.452, 0.996; *p* < 0.005)	7
	Gatto et al., 2021, USA and Canada [120]	AHS-2 cohort	Vegetarian Dietary Patterns	132	58	75.1 ± 8.1	10	↔Cognitive function in the fully adjusted model comparing highest vs. lowest intake			6
	Munoz-Garcia et al., 2020, Spain [37]	SUN cohort study	PVD score: 12–60	806	30.3	61 ± 6	6 ± 3	↔Cognitive function for STICS-m score change in the fully adjusted model comparing highest vs. lowest intake β (95% CI) Q1 0 (Ref) Q2 −0.19 (−0.87, 0.48) Q3 −0.09 (−0.74, 0.56) Q4 0.22 (−0.49, 0.93) Q5 0.41 (−0.56, 1.38) ↔Cognitive function for each 6 points (12–60) in the fully adjusted model β (95% CI; *p* value) 0.19 (−0.03, 0.40; *p* > 0.05)			7

7MS, seven-minute screening; ADG, Australian Dietary Guidelines; AHS-2, Adventist Health Study-2; AIBL, Australian Imaging, Biomarkers and Lifestyle study of ageing; APOE, apolipoprotein E; AusDiab, Australian Diabetes, Obesity and Lifestyle; CHNS, China Health and Nutrition Survey; CI, confidence interval; DDG, Dutch dietary guidelines; DGI, dietary guideline index; DP, dietary pattern; GCF, global cognitive function; GRAS, Geisinger Rural Aging Study; HAICDDS, History-Based Artificial Intelligent Clinical Dementia Diagnostic System; HDI, healthy diet indicator; HDP, healthy dietary pattern; HFDP, healthy food dietary pattern; HR, hazard ratio; IDQ, Index Diet Quality; JDI8, 8-item Japanese Diet Index; LCHP, a low-carbohydrate, high-protein diet; LRGS-TUA, large-scale population-based study among older adults aged 60 years and above in Malaysia; MAS, Sydney Memory and Ageing Study; MCI, mild cognitive impairment; MDCS, Malmö Diet and Cancer study; mDHD15 index, modified Dutch Healthy Diet 2015 index; MMSE, Mini-Mental State Examination; MrOS, Osteoporotic Fractures in Men; n, number; NA, not available; NILS-LSA, National Institute for Longevity Sciences—Longitudinal Study of Aging; NPDP, Nordic prudent dietary pattern; NuAge, Quebec Longitudinal Study on Nutrition and Successful Aging; OR, odds ratio; PATH, Personality and Total Health Through Life Cohort; PD, prudent diet; PDP, prudent dietary pattern; PIVUS, Prospective Investigation of the Vasculature in Uppsala Seniors; PVD, pro-vegetarian diet; Q, quintile; Ref, reference; SD, standard deviation; SDGS, Swedish dietary guidelines score; SE, standard error; SEP, socioeconomic position; SNAC-K, Swedish National study on Aging and Care in Kungsholmen; STICS-m, Spanish Telephone Interview for Cognitive Status; SUN, Seguimiento Universidad de Navarra; T, tertile; TCVS, Tzu Chi Vegetarian Study; TLSA, Taiwan Longitudinal Study of Aging; UK, United Kingdom; USA, United States of America; WELL, Wellbeing Eating and Exercise for a Long Life; WII, Whitehall II study; wJDI9, 9-component-weighted Japanese Diet Index; β, beta coefficient; ↑, significant increase in outcome; ↓, significant decrease in outcome; ↔, no significant effect.

**Table 9 nutrients-17-03469-t009:** Summary of the 12 publications (prospective studies) that investigated associations between WDP and cognitive outcomes.

Author, Year, Region	Study Name	Adherence	Subjects			Study Period (Follow-Up Years)	Outcomes			Study Quality
			Total (*n*)	Female (%)	Age (Range or Mean/SD or Median) (Years)	Average Follow-Up (Year)	Cognitive Function	Cognitive Impairment or MCI	Dementia	
Rogers-Soeder et al., 2024, USA [100]	MrOS	WDP	4231	0	72 ± 5.5	4.6 ± 0.3	↓Cognitive function in the fully adjusted model comparing highest vs. lowest intake ▪3MS scores β (95% CI; *p* value) Q1 (Ref) Q2 −0.09 (−0.16, −0.02; *p* = 0.01) Q3 −0.05 (0.15, −0.08; *p* = 0.13) Q4 −0.01 (−0.08, 0.05; *p* = 0.68) ▪Trail B test time β (95% CI; *p* value) Q1 (Ref) Q2 0.43 (−0.05, 1.00; *p* = 0.08) Q3 0.08 (−0.38, 0.63; *p* = 0.75) Q4 0.3 (−0.19, 0.88; *p* = 0.25)			6
Muñoz-García et al., 2022, Spain [51]	SUN cohort study	WDP	806	30	66 ± 5.5	6	↓Cognitive function for STICS-m score change in the fully adjusted model comparing highest vs. lowest intake β (95% CI) T1 (Ref) T2 −0.49 (−1.03, 0.05) T3 −0.80 (−1.51, −0.08)			7
Samuelsson et al., 2022, Sweden [101]	Gothenburg H70 birth cohort study	WDP	602	64	70.6 ± 0.3	12.8			↑Dementia incidence for APOE ε4 carriers in the fully adjusted model comparing highest vs. lowest intake HR (95% CI; *p* value) 1.37 (1.05, 1.78; *p* = 0.02)	9
Melo van Lent et al., 2022, USA [123]	NHS	EDIP score	16,058	100	74 ± 2	6	↓GCF in the fully adjusted model comparing highest vs. lowest intake β (95% CI) Q1 (Ref) Q2 −0.004 (−0.03, 0.03) Q3 0.01 (−0.02, 0.04) Q4 −0.04 (−0.07, 0.01) Q5 −0.01 (−0.04, 0.02)			7
Parrott et al., 2021, Canada [102]	NuAge Study	WDP	350	54	73.7 ± 3.8	4	↓Cognitive function in the fully adjusted model comparing highest vs. lowest intake ▪Global cognition β (95% CI; *p* value) −0.16 (0.06; *p* = 0.009) ▪Executive function −0.60 (0.27; *p* = 0.027)			6
D’Amico et al., 2020, Canada [124]	NuAge study	WDP	1276	52	74.16 ± 4.16	3	↔Cognitive function in the fully adjusted model comparing highest vs. lowest intake			6
Akbaraly et al., 2019, UK [92]	WII	WDP	8225	30.9	50.2 ± 6.1	24.8	↔Cognitive function for 18 years between per 1-SD increase in WDP in the fully adjusted model β (95% CI; *p* value) −0.01 (−0.04, 0.02; *p* = 0.62)		↔Dementia incidence in the fully adjusted model comparing highest vs. lowest intake ▪WDP in 1991–1993 HR (95% CI) T1 1 (Ref) T2 0.86 (0.64, 1.16) T3 1.00 (0.70, 1.43) Per 1-SD (10-point) in increase: 0.99 (0.83, 1.17) ▪WDP in 1997–1999 T1 1 (Ref) T2 0.80 (0.53, 1.19) T3 0.96 (0.60, 1.54) Per 1-SD (10-point) in increase: 1.03 (0.82, 1.30) ▪WDP in 2002–2004 T1 1 (Ref) T2 0.81 (0.55, 1.19) T3 0.80 (0.50, 1.28) Per 1-SD (10-point) in increase: 0.89 (0.71, 1.12)	7
Dearborn-Tomazos et al., 2019, USA [125]	ARIC study	WDP	13,588	55.9	54.6 ± 5.7	20	↔Cognitive function in the fully adjusted model comparing highest vs. lowest intake			8
Shakersain et al., 2016, Sweden [103]	SNAC-K	WDP	2223	60.8	70.6 ± 8.9	6	↓Cognitive function for MMSE in the fully adjusted model comparing highest vs. lowest intake β (95% CI; *p* value) −0.156 (−0.24, −0.073; *p* < 0.001)			8
Gardener et al., 2015, Australia [72]	AIBL study	WD score	527	60.2	69.3 ± 6.4	3	↓Visuospatial functioning for APOE ε4 allele carriers ↔Cognitive decline in the fully adjusted model comparing highest vs. lowest intake			6
Tsai et al., 2015, Taiwan [104]	TLSA study	WDP	2988	45.7	73 ± 6	8	↓Cognitive function in the fully adjusted model comparing highest vs. lowest intake OR (95% CI) 4.35 (1.52, 12.50)			7
Parrott et al., 2013, Canada [105]	NuAge study	WDP	1099	49.4	74.1 ± 4.1	3	↓Cognitive function WDP with low education comparing highest vs. lowest intake β (95% CI; *p* value) −0.23 (−0.43, −0.032; *p* = 0.023)			6

3MS, Modified Mini-Mental State Examination; AIBL, Australian Imaging, Biomarkers and Lifestyle study of ageing; APOE, apolipoprotein E; ARIC, Atherosclerosis Risk in Communities; CI, confidence interval; EDIP, empirical dietary inflammatory pattern; GCF, global cognitive function; HR, hazard ratio; MCI, mild cognitive impairment; MMSE, Mini-Mental State Examination; MrOS, Osteoporotic Fractures in Men; n, number; NHS, Nurses’ Health Study; NuAge, Quebec Longitudinal Study on Nutrition and Successful Aging; OR, odds ratio; Q, quintile; Ref, reference; SD, standard deviation; SNAC-K, Swedish National study on Aging and Care in Kungsholmen; STICS-m, Spanish Telephone Interview for Cognitive Status; SUN, Seguimiento Universidad de Navarra; T, tertile; TLSA, Taiwan Longitudinal Study of Aging; UK, United Kingdom; USA, United States of America; WD, Western diet; WDP, Western dietary pattern; WII, Whitehall II study; β, beta coefficient; ↑, significant increase in outcome; ↓, significant decrease in outcome; ↔, no significant effect.

**Table 10 nutrients-17-03469-t010:** Summary of the 13 publications (16 prospective studies) that investigated associations between other dietary patterns and cognitive outcomes.

Dietary Pattern	Author, Year, Region	Study Name	Adherence	Subjects			Study Period (Follow-Up Years)	Outcomes			Study Quality
				Total (*n*)	Female (%)	Age (Range or Mean/SD or Median) (Years)	Average Follow-Up (Year)	Cognitive Function	Cognitive Impairment or MCI	Dementia	
Animal-based patterns	Hu et al., 2023, China [126]	CLHLS	ADI	17,827	53.24	86.35 ± 10.20	1998–2018		↑Cognitive impairment in the fully adjusted model comparing highest vs. lowest intake HR (95% CI; *p* value) T1 1 (Ref) T2 1.35 (1.19, 1.53; *p* < 0.001) T3 1.64 (1.38, 1.96; *p* < 0.001)		7
	Chen et al., 2017, Taiwan [114]	NA (elderly health checkup program at National Taiwan University Hospital, Taipei, Taiwan)	Meat dietary pattern	475	52	≥65	2	↓Cognitive function for verbal fluency—total score and digit span—reverse score in the fully adjusted model comparing highest vs. lowest intake ▪↓Verbal fluency—total score β (95% CI) T1 (Ref) T2 −0.10 (−0.24, 0.04) T3 −0.19 (−0.35, −0.02) ▪↓Digit span—reverse β (95% CI) T1 (Ref) T2 0.20 (0.04, 0.36) T3 0.22 (0.04, 0.41) ↔Global cognition in the fully adjusted model comparing highest vs. lowest intake			5
	Tomata et al., 2016, Japan [109]	Ohsaki Cohort 2006 Study	Animal dietary pattern	14,402	56	73.8 ± 5.9	4.9 ± 1.5			↔Dementia incidence in the fully adjusted model comparing highest vs. lowest intake HR (95% CI) Q1 1 (Ref) Q2 1.09 (0.93, 1.28) Q3 1.13 (0.95, 1.33) Q4 1.12 (0.92, 1.36)	8
	Titova et al., 2013, Sweden [82]	PIVUS	Meat & meat products	194	50	70.1 ± 0.01	5	↓Cognitive function for 7MS in the fully adjusted model comparing highest vs. lowest intake β (*p* value) −0.26 (*p* < 0.001)			8
Sugar dietary pattern	Zhang et al., 2024, UK [127]	UK Biobank study	High-sugar dietary score	210,832	55	56.08 ± 7.99	11.80 ± 1.66			↑All-cause dementia risk in the fully adjusted model comparing highest vs. lowest intake HR (95% CI) Q1 (Ref) Q2 0.914 (0.778, 1.074) Q3 0.964 (0.822, 1.132) Q4 1.255 (1.078, 1.462)	9
Poor dietary pattern	Xu et al., 2023, UK [128]	UK Biobank study	Poor dietary pattern	497,533	54.4	56.5 ± 8.1	14.8			↔Dementia HR (95% CI) 1.04 (0.99, 1.09)	8
Iron-related dietary pattern	Shi et al., 2019, China [129]	CHNS	Iron-related dietary pattern	4852	52	≥55	16	↔Cognitive function in the fully adjusted model comparing highest vs. lowest intake OR (95% CI) Q1 1 (Ref) Q2 1.06 (0.86, 1.30) Q3 1.24 (0.99, 1.54) Q4 1.50 (1.17, 1.93)			9
Traditional dietary pattern	Corley et al., 2021, Scotland [55]	Lothian Birth Cohort 1936 study	Traditional dietary pattern	863	49.7	69.5 ± 0.8	12.5 ± 0.5	↔Cognitive function in the fully adjusted model			8
Traditional Chinese dietary pattern	Xu et al., 2018, China [130]	CHNS	Traditional Chinese dietary pattern (heavily on rice, pork, and fish, and inversely on wheat and whole grain)	4847	52	≥55	10	↑Cognitive function in the fully adjusted model comparing highest vs. lowest intake β (95% CI) ▪Global score Q1 1 (Ref) Q2 1.10 (0.70, 1.50) Q3 0.86 (0.43, 1.28) Q4 1.32 (0.90, 1.73)			8
Protein-rich dietary pattern			Protein-rich dietary pattern					↑Cognitive function in the fully adjusted model comparing highest vs. lowest intake β (95% CI) ▪Global score Q1 1 (Ref) Q2 0.72 (0.32, 1.12) Q3 1.66 (1.24, 2.08) Q4 2.28 (1.80, 2.76) ▪Verbal memory score Q1 1 (Ref) Q2 0.41 (0.12, 0.70) Q3 0.99 (0.69, 1.30) Q4 1.36 (1.01, 1.71)			
Starch-rich dietary pattern			Starch-rich dietary pattern (high intake of salted vegetables, legumes, whole grain, and tubers)					↓Cognitive function in the fully adjusted model comparing highest vs. lowest intake β (95% CI) ▪Global score Q1 1 (Ref) Q2 −0.20 (−0.57, 0.18) Q3 −0.12 (−0.50, 0.26) Q4 −0.31 (−0.70, 0.08) ▪Verbal memory score Q1 1 (Ref) Q2 −0.17 (−0.44, 0.10) Q3 −0.22 (−0.49, 0.05) Q4 −0.43 (−0.71, −0.15)			
Traditional Chinese dietary pattern	Chen et al., 2017, Taiwan [114]	NA (elderly health checkup program at National Taiwan University Hospital, Taipei, Taiwan)	Traditional Chinese dietary pattern (pickled vegetables and fermented foods)	475	52	≥65	2	↑Cognitive function for Logical Memory-Recall I in the fully adjusted model comparing highest vs. lowest intake ▪↑Logical Memory-Recall I β (95% CI) T1 (Ref) T2 0.06 (0.10, 0.21) T3 0.18 (0.02, 0.33) ↔Global cognition in the fully adjusted model comparing highest vs. lowest intake			5
Legumes pattern	Mazza et al., 2017, Italy [131]	NA	Legumes pattern	214	NA	70 ± 4	1	↑Cognitive function for MMSE and ADAS-cog in the fully adjusted model comparing highest vs. lowest intake β (95% CI) ▪MMSE 0.25 (0.07, 0.44) ▪ADAS-cog −0.10 (−0.79, −0.30)			5
Taiwan’s traditional dietary pattern	Tsai et al., 2015, Taiwan [104]	TLSA study	Taiwan’s traditional dietary pattern (more soy, rice, wheat, and salt but less meat and milk products than WDP)	2988	45.7	73 ± 6	8	↔Cognitive function in the fully adjusted model comparing highest vs. lowest intake OR (95% CI) 1.37 (0.85, 2.21)			7
Inflammatory dietary pattern	Ozawa et al., 2017, UK [132]	WII	IDP	5083	28.7	45–79	10	↓Reasoning in the fully adjusted model comparing highest vs. lowest intake T1 −0.31 (−0.34, −0.28) T2 −0.36 (−0.39, −0.33) T3 −0.37 (−0.40, −0.33) ↓Global cognition in the fully adjusted model comparing highest vs. lowest intake T1 −0.31 (−0.33, −0.28) T2 −0.35 (−0.37, −0.32) T3 −0.35 (−0.38, −0.32)			7
High dairy dietary pattern	Tomata et al., 2016, Japan [109]	Ohsaki Cohort 2006 Study	High dairy dietary pattern	14,402	56	73.8 ± 5.9	4.9 ± 1.5			↔Dementia incidence in the fully adjusted model comparing highest vs. lowest intake HR (95% CI) Q1 1 (Ref) Q2 0.88 (0.76, 1.03) Q3 0.99 (0.84, 1.16) Q4 0.97 (0.83, 1.15)	8
Fish, legumes, and vegetables pattern	Ashby-Mitchell et al., 2015, Australia [115]	AusDiab study	Fish, legumes, and vegetables pattern	577	49.22	66.07 ± 4.85	3		↓Cognitive impairment in the fully adjusted model comparing highest vs. lowest intake OR (95% CI; *p* value) 1.032 (1.001, 1.064; *p* = 0.04)		6
Dairy, cereal, and eggs pattern	Ashby-Mitchell et al., 2015, Australia [115]	AusDiab study	Dairy, cereal, and eggs pattern	577	49.22	66.07 ± 4.85	3		↓Cognitive impairment in the fully adjusted model comparing highest vs. lowest intake 1.020 (1.007, 1.033; *p* = 0.003)		6

7MS, seven-minute screening; ADAS-Cog, Alzheimer’s disease assessment scale—cognitive sub-scale; ADI, animal-based diet index; AusDiab, Australian Diabetes, Obesity and Lifestyle; CHNS, China Health and Nutrition Survey; CI, confidence interval; CLHLS, China Longitudinal Healthy Longevity Survey; HR, hazard ratio; IDP, inflammatory dietary pattern; MCI, mild cognitive impairment; MMSE, Mini-Mental State Examination; n, number; NA, not available; OR, odds ratio; PIVUS, Prospective Investigation of the Vasculature in Uppsala Seniors; Q, quartile; Ref, reference; SD, standard deviation; T, tertile; TLSA, Taiwan Longitudinal Study of Aging; UK, United Kingdom; WII, Whitehall II study; β, beta coefficient; ↑, significant increase in outcome; ↓, significant decrease in outcome; ↔, no significant effect.

## Data Availability

No new data were created or analyzed in this study. Data sharing is not applicable to this article.

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
