# Peer review of "Impact of Diverse Dietary Patterns on Cognitive Health: Cumulative Evidence from Prospective Cohort Studies"

_nutrients, 2025, doi:10.3390/nu17213469_

Round 1

Reviewer 1 Report

Comments and Suggestions for Authors

The authors have conducted a systematic review of the impact of dietary pattern on cognitive impairment/dementia. The review appears comprehensive and the manuscript contributes important evidence to the research field.

Comments:

Systematic review protocols should be registered for reasons of integrity and transparency. PROSPERO is an example: https://www.crd.york.ac.uk/prospero/

Did the authors register their review?

Lines 60-61: This is a duplicate figure legend.

Search terms should be listed.

Listing inclusion/exclusion criteria in a table format would be easier to read.

Tables: Add "years" to column heading for age.

Lines 511- : It would be helpful for some readers to clarify that the Western dietary pattern is considered unhealthful. Related to this, please add to the discussion a paragraph or more about the general findings of the impact of unhealthful dietary patterns (Western, sugar, poor) on cognitive outcomes. Discuss the mechanisms by which unhealthful diets may negatively impact cognitive function with aging. 

Comments on the Quality of English Language

English editing is needed. Below are a few examples.

Line 658: Incomplete sentence. 

667, 673: The structured group.

624-627 and 645-656: Avoid one-sentence paragraphs. Please write fully developed paragraphs. 

Author Response

Comments and Suggestions for Authors

The authors have conducted a systematic review of the impact of dietary pattern on cognitive impairment/dementia. The review appears comprehensive and the manuscript contributes important evidence to the research field.

Comments:

Systematic review protocols should be registered for reasons of integrity and transparency. PROSPERO is an example: https://www.crd.york.ac.uk/prospero/

Did the authors register their review?

Thank you for your comment. This article has the characteristics of a comprehensive and updated review combined with a systematic review and a narrative review. Therefore, we have now revised the title to “Impact of Diverse Dietary Patterns on Cognitive Health: Cumulating Evidence from Prospective Cohort Studies”. We also have addressed more detailed potential mechanisms of the association between different dietary patterns and cognitive outcomes.

Lines 60-61: This is a duplicate figure legend.

Thank you for your comment. It has been addressed in line 71.

Search terms should be listed.

Thank you for your comment. It has been addressed in lines 79-83.

Listing inclusion/exclusion criteria in a table format would be easier to read.

Thank you for your comment. It has been addressed in lines 94-98.

Tables: Add "years" to column heading for age.

Thank you for your comment. We have added “years” to the column heading for age in Tables 3-10 to clarify the units.

Lines 511- : It would be helpful for some readers to clarify that the Western dietary pattern is considered unhealthful. Related to this, please add to the discussion a paragraph or more about the general findings of the impact of unhealthful dietary patterns (Western, sugar, poor) on cognitive outcomes. Discuss the mechanisms by which unhealthful diets may negatively impact cognitive function with aging.

Thank you. We have added about the general findings of the impact of unhealthful dietary patterns (Western, sugar, poor) on cognitive outcomes. In addition, discussion on the mechanisms by which unhealthful diets may negatively impact cognitive function with aging has been added. Please refer to the lines from 722-750 for the comments.

Comments on the Quality of English Language

English editing is needed. Below are a few examples.

Thank you. We have correctly edited through the texts after carefully speculate our manuscript. 

Line 658: Incomplete sentence.

Thank you. We have deleted the sentences because they were not appropriate in the order of the flow of context.

The grammar correct sentences regarding the mechanisms of the association between the WDP and cognitive outcomes have been described in lines 722-732.

667, 673: The structured group.

Thank you. We have divided paragraphs based on the flow of context and the structure of the sentences.

624-627 and 645-656: Avoid one-sentence paragraphs. Please write fully developed paragraphs.

Thank you. We have revised and divided paragraphs clarifying the flow of context and the structure of the sentences.

Reviewer 2 Report

Comments and Suggestions for Authors

Dear Authors,

Congratulations on tackling an issue important for health culture and the challenges of an aging society. I have a few comments on the paper.

1) For a more holistic approach, the introduction could have included a broader discussion of other lifestyle-related, modifiable risk factors for dementia, and then focused on the key dietary factors.

2) I suggest formulating and developing the paper's purpose more precisely.

3) I feel that not all the dietary patterns analyzed were sufficiently discussed and interpreted in the discussion regarding potential mechanisms of influence on cognitive function, its impairment, and the development of dementia.

4) The conclusions are too concise and need further consideration and refinement.

Author Response

Comments and Suggestions for Authors

Dear Authors,

Congratulations on tackling an issue important for health culture and the challenges of an aging society. I have a few comments on the paper.

1) For a more holistic approach, the introduction could have included a broader discussion of other lifestyle-related, modifiable risk factors for dementia, and then focused on the key dietary factors.

Thank you for your suggestions. We have added a description of the results of other risk factors for dementia in the introduction section. Please see the lines 43-49.

2) I suggest formulating and developing the paper's purpose more precisely.

Thank you for your comments. We have added a specific descriptive sentence about the purpose of the review to the end of the introduction. Please check the lines 67-69.

3) I feel that not all the dietary patterns analyzed were sufficiently discussed and interpreted in the discussion regarding potential mechanisms of influence on cognitive function, its impairment, and the development of dementia.

Thank you. We have sufficiently discussed and interpreted regarding the potential mechanisms of influence on cognitive function, its impairment, and the development of dementia in lines 722-750.

4) The conclusions are too concise and need further consideration and refinement.

Thank you. We have revised our conclusion with further consideration and refinement in abstract and discussion.

Reviewer 3 Report

Comments and Suggestions for Authors

This review gives a comprehensive description of all different dietary patterns that may effect cognitive function by prevention or reducing cognitive decline. Interestingly, but not surprisingly it was concluded that the MIND diet and hPDI appeared was most effective.

Comments on the Quality of English Language

Check the language once again, some prepositions has to be checked and so on.

Author Response

Comments and Suggestions for Authors

This review gives a comprehensive description of all different dietary patterns that may affect cognitive function by prevention or reducing cognitive decline. Interestingly, but not surprisingly it was concluded that the MIND diet and hPDI appeared was most effective.

Thank you.

Check the language once again, some prepositions has to be checked and so on.omments on the Quality of English Language.

Thank you for your comments. We have revised and edited through the texts after looking at the manuscript.   Now, this manuscript has the high quality of English Language.

Round 2

Reviewer 1 Report

Comments and Suggestions for Authors

The authors have revised the manuscript appropriately in response to my comments. 

Minor revisions still needed:

Line 49: "...can have a negative impact on dementia" change to "negative impact on cognitive function."

The PRISMA flow diagram would be better positioned after search terms and inclusion/exclusion criteria descriptions. 

Line 775: "The WDP is associated with the decreased risk of cognitive function." This is incorrect. It should read, "The WDP is associated with the increased risk of cognitive function."

Comments on the Quality of English Language

Minor errors are noted above.

Author Response

Comments and Suggestions for Authors

The authors have revised the manuscript appropriately in response to my comments. 

Minor revisions still needed:

Line 49: "...can have a negative impact on dementia" change to "negative impact on cognitive function."

Thank you for your comment. It has been addressed in line 49.

The PRISMA flow diagram would be better positioned after search terms and inclusion/exclusion criteria descriptions. 

Thank you for your comment. It has been addressed in lines 109-111.

Line 775: "The WDP is associated with the decreased risk of cognitive function." This is incorrect. It should read, "The WDP is associated with the increased risk of cognitive function."

Thank you for your comment. It has been addressed in line 779.

Reviewer 2 Report

Comments and Suggestions for Authors

Thank you for taking my comments into account. 

Author Response

Comments and Suggestions for Authors

 Thank you for taking my comments into account.

Thank you.